# CURVATURE DIVERSITY-DRIVEN DEFORMATION AND DOMAIN ALIGNMENT FOR POINT CLOUD

## ABSTRACT

Unsupervised Domain Adaptation (UDA) is crucial for reducing the need for extensive manual data annotation when training deep networks on point cloud data. A significant challenge of UDA lies in effectively bridging the domain gap. To tackle this challenge, we propose **C**urvature **D**iversity-Driven **N**uclear-Norm Wasserstein **D**omain Alignment (CDND). Our approach first introduces a ***Curv**ature Diversity-driven Deformation **Rec**onstruction (CurvRec)* task, which effectively mitigates the gap between the source and target domains by enabling the model to extract salient features from semantically rich regions of a given point cloud. We then propose *Deformation-based Nuclear-norm Wasserstein Discrepancy (D-NWD)*, which applies the Nuclear-norm Wasserstein Discrepancy to both *deformed and original* data samples to align the source and target domains. Furthermore, we contribute a theoretical justification for the effectiveness of D-NWD in distribution alignment and demonstrate that it is *generic* enough to be applied to **any** deformations. To validate our method, we conduct extensive experiments on two public domain adaptation datasets for point cloud classification and segmentation tasks. Empirical experiment results show that our CDND achieves state-of-the-art performance by a noticeable margin over existing approaches.

## 1 INTRODUCTION

Adopting deep neural network on point cloud representation learning has led to significant success in various applications, including robotics Maturana & Scherer (2015); Duan et al. (2021), autonomous vehicles Mahjourian et al. (2018); Cui et al. (2021), and scene understanding Zheng et al. (2013); Zhu et al. (2017). Most works rely on supervised learning Su et al. (2015); Wu et al. (2015); Qi et al. (2017) and assume that training and testing data are sampled from the same distribution. However, acquiring labels for training data is both time-consuming and labor-intensive. Moreover, testing data may be from a different distribution w.r.t. training data in real-world scenarios, known as '*domain gap*'. Unsupervised domain adaptation (UDA) offers a solution to tackle these issues by utilizing knowledge transfer from source domains with annotated data to target domains with only unlabeled data. Although UDA is well studied for 2D planner data, e.g., images, UDA for 3D point clouds has not been explored extensively due to challenges such as irregular, unstructured, and unordered nature of 3D point cloud data. Such irregularities exacerbate geometric variations between the source and target domains compared to the 2D planner data and make extending existing solutions nontrivial.

To address the above challenges, we propose **C**urvature **D**iversity-Driven **N**uclear-Norm Wasserstein **D**omain Alignment (CDND). Our first contribution is a deformation reconstruction method that leverages *curvature diversity* in different regions of a point cloud for domain alignment. We evaluate curvature diversity based on the entropy that captures the saliency of each region. This metric is then used to select regions for deformation and reconstruction. Unlike previous methods such as Achituve et al. (2021), our approach strategically selects regions based on their information content. Also, distinct from Zou et al. (2021), which selects regions with high curvature and classifies them into a fixed set, we select regions with low curvature diversity and focus on deforming and reconstructing these regions rather than classifying them. Our method avoids deforming semantically rich regions and enables the feature extractor to focus on extracting features from these regions. Our second contribution is the **D**eformation-based **N**uclear-norm **W**asserstein **D**iscrepancy (D-NWD). Unlike NWD, D-NWD incorporates features from both original and deformed samples when aligning the source and target domains. While including features from deformed samples cre-

ates a diverse and robust feature space that may improve model generalization under domain shift, our primary contribution lies in the theoretical analysis of D-NWD. This analysis demonstrates that D-NWD can reduce the domain gap between the source and target domains, showcasing its effectiveness. Our analysis also illustrates that D-NWD is *generic* enough to be used for any deformation method, not just the one presented in this paper. Experiments on common benchmarks for both classification and segmentation show that our approach achieves state-of-the-art performance.

## 2 RELATED WORKS

**Domain Adaptation on Point Clouds.** Despite extensive works on UDA for 2D planner image data Ganin & Lempitsky (2015); Tzeng et al. (2017); Mansour et al. (2008), only a limited number of studies Qin et al. (2019); Achituve et al. (2021); Shen et al. (2022); Zou et al. (2021) address UDA in point clouds and non-planner data spaces as extending methods for 2D data for point clouds in nontrivial. Qin *et al*. Qin et al. (2019) introduce PointDAN that integrates both local and global domain alignment strategies. They also provide the PointDA benchmark for point cloud classification under the UDA setting. Achituve *et al*. Achituve et al. (2021) propose a domain alignment technique that involves reconstruction from deformation and incorporates PointMixup Chen et al. (2020). They also introduce the PointSegDA benchmark for point cloud segmentation under the UDA setting. Zou *et al*. Zou et al. (2021) utilize two geometry-inspired self-supervised classification tasks to learn domain-invariant feature. Shen *et al*. Shen et al. (2022) introduce a self-supervised method for learning geometry-aware implicit functions to handle domain-specific variations effectively. Our work differs from these approaches by proposing more sophisticated self-supervised learning tasks and a generic theoretical framework, leading to state-of-the-art performance.

**Optimal Transport for Domain Adaptation.** The Wasserstein metric, known for encoding the natural geometry of probability measures within optimal transport theory, has been extensively studied for its application in domain adaptation due to its nice properties. Gautheron *et al*. Gautheron et al. (2019) propose Wasserstein Distance Guided Representation Learning to leverage the Wasserstein distance to enhance similarities between embedded features. Lee et al. (2019) and Gabourie et al. (2019) propose to use the sliced Wasserstein discrepancy instead of $L_1$ distance in Maximum Classifier Discrepancy Saito et al. (2018) to achieve a more geometrically meaningful intra-class divergence. Additionally, CGDM Du et al. (2021) introduces cross-domain gradient discrepancy to further mitigate domain differences. DeepJ-DOT Damodaran et al. (2018) utilizes a coupling matrix to map source samples to the target domain. Gautheron *et al*. Gautheron et al. (2019) propose a feature selection technique that addresses the domain shifts problem. Moreover, Xu *et al*. Xu et al. (2020) develop reliable weighted optimal transport, which uses spatial prototypical information and intra-domain structure to evaluate sample-level domain discrepancies, resulting in a better pairwise optimal transport plan. Finally, Fatras *et al*. Fatras et al. (2021) present an unbalanced optimal transport method combined with a mini-batch strategy to efficiently learn from large-scale datasets. In this work, we developed our D-NWD based on another previous method, NWD Chen et al. (2022).

## 3 PROPOSED METHOD

We begin by defining the unsupervised domain adaptation problem and then we provide an overview of our UDA approach, called **C**urvature **D**iversity-Driven **N**uclear-Norm Wasserstein **D**omain Alignment (CDND), in Section 3.1. Next, we detail our main contributions: (1) the **Curv**ature Diversity-based Deformation **Rec**onstruction method (CurvRec), as discussed in Section 3.2 and Section 3.3, and (2) the **D**eformation-based **N**uclear-norm **W**asserstein **D**iscrepancy (D-NWD) in Section 3.4. Following in Section 4, we present our theoretical contribution of D-NWD.

### 3.1 PROBLEM FORMULATION

We consider a source domain with labeled samples and a target domain, differing from the source, with unlabeled samples. Our goal is to develop a UDA model to accurately predict labels for the target domain using both the source labeled dataset and the target unlabeled dataset. Let $\mathcal{S}$ represent the source domain, where $X_s^i$ denotes the $i$-th batch of samples and $y_s^i$ their corresponding labels. Similarly, let $\mathcal{T}$ represent the target domain, where $X_t^i$ is the $i$-th batch of samples. The feature space induced by $\mathcal{S}$ and $\mathcal{T}$ is denoted by $\Omega_o$. In addition, we introduce deformed domains $\mathcal{S}^d$ and

$\mathcal{T}^d$, with their feature space $\Omega_d$. We assume $\Omega_o$ and $\Omega_d$ to be disjoint and $\Omega_o \cup \Omega_d \subseteq \mathbb{R}^n$. A point cloud from the source domain is denoted as $x_s \in \mathbb{R}^{n \times 3}$ and from the target domain is $x_t \in \mathbb{R}^{n \times 3}$, where $n$ is the number of points. The corresponding deformed point clouds are denoted by $x_s^d$ and $x_t^d$, respectively.

The pipeline of our CDND is presented in Figure 1. Our model first uses a feature extractor $E$ to obtain shape features from both source and target point clouds. To minimize domain gaps and ensure domain-invariant features, we: (1) use a curvature diversity-driven deformation reconstruction task using a reconstruction decoder $h_{\text{SSL}}$ and (2) employ the D-NWD to align domains through a classifier $C$. The aligned features are then used for downstream tasks, *i.e.*, cloud classification and segmentation. The model is trained using source-labeled and target-unlabeled data.

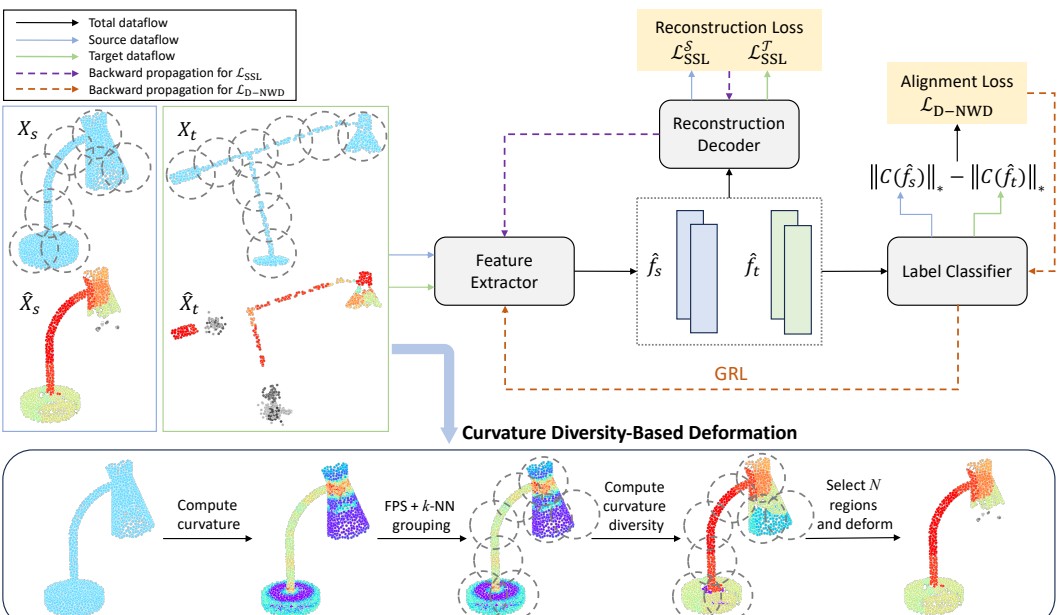

Figure 1: Pipeline of CDND. The inputs are the source batch $X_s$ and target batch $X_t$. We first deform them into $\hat{X}_s$ and $\hat{X}_t$ using Curvature Diversity-Based Deformation. Next, $X_s$, $X_t$, $\hat{X}_s$, and $\hat{X}_t$ are sent into a feature extractor. The features of deformed samples are fed into a reconstruction decoder to reconstruct the deformed regions. For domain alignment, both original and deformed features are sent to D-NWD. Aside from the two losses shown in the figure, a cross-entropy loss is computed on $X_s$ and $\hat{X}_s$ with labels. An NWD loss $\mathcal{L}_{\text{NWD}}^{\mathcal{T}}$ on $X_t$ and $\hat{X}_t$ is also computed to ensure prediction consistency between the target original and deformed pairs.

## 3.2 CURVATURE DIVERSITY-DRIVEN DEFORMATION

To extract domain-invariant features shared by both source and target domains, Achituve *et al.* proposed *deformation reconstruction* Achituve et al. (2021). Specifically, there are three deformation strategies introduced in Achituve et al. (2021) for deforming point clouds, *i.e.*, volume-based, feature-based, and sample-based, according to the way of dividing point clouds into regions for deformation.

Although the strategies mentioned above use different techniques to select regions for deformation, they all randomly divide a point cloud into regions and uniformly select regions based on their spatial locations or arrangements. However, this approach may not be optimal as regions within a point cloud vary in their semantic richness, *i.e.*, some regions contain more semantic information. These semantically rich regions are crucial for tasks such as classification, as they have more distinguishable characteristics. For instance, to differentiate a point cloud of a plant from that of a lamp, focusing on the leaves and flowers — which have richer semantic information — would be more effective than focusing on the flower pot, which is similar to the base of a lamp. Thus, deforming regions with richer semantic information causes the point cloud to lose semantic meaning, making

it difficult for a classifier to classify it. To encourage the feature extractor to prioritize regions with rich information, we propose deforming regions that are *less* semantically rich. This strategy helps learning to extract features from the most informative or salient regions of a point cloud.

To evaluate the richness of semantics, we propose using curvature diversity as a measurement. Following Zou *et al.* Zou et al. (2021), we compute point cloud curvature using PCA Abdi & Williams (2010). Specifically, we first select a small neighborhood around each point and apply PCA to determine the principal directions and their eigenvalues. The curvature is then calculated as:

$$c = \frac{|\lambda_{\min}|}{\sum_{i=1}^{K} |\lambda_i|} \tag{1}$$

where $\lambda_{\min}$ is the smallest eigenvalue of the matrix, and $K$ is the number of eigenvalues. Larger variation in curvature indicates a more intricate geometry and more significant shape changes within a region. The fourth lamp sample in the bottom row of Figure 1 illustrates this property: regions with warmer colors represent areas of higher curvature diversity. To measure the diversity or variation of curvature in a region, we propose to use *entropy of curvature*. Entropy effectively captures the variability and complexity of the curvature, allowing us to quantify the richness of semantics within a region. Formally, we use the following measure the curvature diversity:

$$c_{\min}^{j} = \min_{c_i^j \in R^j} \{c_i^j\}_{i=1}^{N_{R^j}}, \quad c_{\max}^{j} = \max_{c_i^j \in R^j} \{c_i^j\}_{i=1}^{N_{R^j}}$$

$$c_{i,\text{norm}}^{j} = \frac{c_i^j - c_{\min}^j}{c_{\max}^j - c_{\min}^j + 1 \times 10^{-10}}, \quad H(c_{\text{norm}}^j) = -\sum_{i=1}^{N_{R^j}} c_{i,\text{norm}}^j \cdot \log(c_{i,\text{norm}}^j + 1 \times 10^{-10}) \tag{2}$$

where, $c_i^j$ represents the curvatures of the $i$-th point in the $j$-th region of the point cloud which contains $N_{R^j}$ points in total. To standardize these values, we first calculate $c_{\min}^j$ and $c_{\max}^j$, which are the minimum and maximum values of all curvatures within a region, respectively. Using these values, we then normalize the curvature values to be in [0, 1], denoted as $\{c_{i,\text{norm}}^j\}$. Then, we calculate the curvature diversity $H(c_{\text{norm}}^j)$ by applying entropy.[1]

For the curvature diversity-driven deformation, we adopt the following steps. First, we use Farthest Point Sampling (FPS) Moenning & Dodgson (2003) to sample $k$ points as centers of $k$ regions. Then, for each center point, we use $k$-Nearest Neighbor ($k$-NN) to select $m$ nearest points, *i.e.*, each region is formed by a center point along with these $m$ nearest points. Next, we select the $N$ regions with the *smallest* curvature diversity to deform. To deform these selected regions, we replace all the points within these regions with new points. These new points are generated by sampling from a Gaussian distribution, where the mean is set to the average position of all the original points in that region, and the variance is set to 0.001. In Figure 1, $\hat{X}_s$ and $\hat{X}_t$ represent the deformed samples, and the points shown in grayscale are those drawn from the Gaussian distribution.

### 3.3 DEFORMATION RECONSTRUCTION LOSS

After deforming the selected regions, we obtain a deformed point cloud $x^d$ from the original $x$. The deformed input $x^d$ is processed by the feature extractor $E$ to generate $E(x^d)$, which is then passed to a reconstruction decoder $h_{\text{SSL}}$ to reconstruct $x$. The self-supervised loss $\mathcal{L}_{\text{SSL}}$ minimizes the distance between $h_{\text{SSL}}(E(x^d))$ and $x$. We use the Chamfer distance in $\mathcal{L}_{\text{SSL}}$, focusing on the original points in $x$ within the deformed region $R$ and their reconstructions from $x^d$. Formally, let $I \subset \{1, 2, \ldots, m\}$ represent the indices of the points in $x \cap R$, and we define $\mathcal{L}_{\text{SSL}}$ as:

$$\mathcal{L}_{\text{SSL}} = \sum_{(x^d, x) \in \mathcal{S} \cup \mathcal{T}} D\left(\{x_i\}_{i \in I}, \{h_{\text{SSL}}(E(x^d))_i\}_{i \in I}\right), \tag{3}$$

where $x_i$ is the $i$-th point in the point cloud $x$ and the Chamfer distance $D$ is defined as :

$$D(R_1, R_2) = \sum_{a \in R_1} \min_{b \in R_2} \|a - b\|_2^2 + \sum_{b \in R_2} \min_{a \in R_1} \|b - a\|_2^2, \tag{4}$$

where $D(R_1, R_2)$ measures the discrepancy between point cloud regions $R_1, R_2 \subset \mathbb{R}^3$. Note that we reconstruct only the deformed regions to reduce computational resources and time.

---

[1]Rigorously speaking, $\{c_{i,\text{norm}}^j\}$ is an un-normalized distribution without being divided by a partition function or normalization constant, but it does not affect our claim of curvature diversity.

## 3.4 DOMAIN ALIGNMENT VIA D-NWD

The curvature diversity-driven deformation reconstruction helps reduce the domain gap between the source and target domains. To further complete classification or segmentation tasks in the presence of domain gap, we propose D-NWD align domains, as inspired by the Nuclear-norm Wasserstein discrepancy (NWD) Chen et al. (2022). A brief overview of NWD is provided in Appendix A. Our D-NWD objective is defined as:

$$W_N(\nu_{s\cup s^d}, \nu_{t\cup t^d}) = \sup_{\|\|C\|_*\|_L \leqslant K} \mathbb{E}_{\hat{f}_s \sim \nu_{s\cup s^d}}[\|C(\hat{f}_s)\|_*] - \mathbb{E}_{\hat{f}_t \sim \nu_{t\cup t^d}}[\|C(\hat{f}_t)\|_*] \tag{5}$$

where $K$ is the Lipschitz constant. Here, $\nu_{s\cup s^d}$ and $\nu_{t\cup t^d}$ are probability measures defined over $\Omega_o \cup \Omega_d$, for the features from samples in original and deformed source and target domains. We align the probability measure of features from original and deformed samples in the source domain with that of the target domain. Our motivation is that taking features from deformed samples into account would provide a richer, more robust feature space, reduce overfitting, and increase the model's adaptability to variations inherent in real-world data. This differs from NWD, which aligns $\nu_s$ and $\nu_t$ defined over $\Omega_o$, the probability measures for the features from samples in original source and target domains. Empirically, our objective in Eq. 5 be approximated by $\hat{\mathcal{L}}_{\text{D-NWD}}$:

$$\mathcal{L}_{\text{D-NWD}} = \frac{1}{N_s} \sum_{i=1}^{N_s} \|C(\hat{f}_s^i)\|_* - \frac{1}{N_t} \sum_{i=1}^{N_t} \|C(\hat{f}_t^i)\|_* \tag{6}$$

where $C$ denotes the classifier, and $\|\cdot\|_*$ represents the nuclear norm. $\hat{f}_s^i \sim \nu_{s\cup s^d}$ represents the features for the $i$-th source batch and $\hat{f}_t^i \sim \nu_{t\cup t^d}$ represents the features for the $i$-th target batch. The ratio between the original samples and deformed samples is 1:1. In practice, we obtain the original and deformed samples by first sampling from the original domain, and then generating the corresponding deformed versions. The alignment is then performed through a min-max game as:

$$\min_E \max_C \mathcal{L}_{\text{D-NWD}} \tag{7}$$

To avoid alternating updates, we employ a Gradient Reverse Layer Ganin et al. (2016), following the approach in Chen et al. (2022), to make the learned features discriminative and domain-agnostic.

## 3.5 OVERALL LOSS

In addition to deformation and domain alignment loss defined in Eq. 3 and Eq. 7, we use a cross-entropy loss $\mathcal{L}_{\text{CLS}}$ on both original and deformed source domain samples for supervised training:

$$\mathcal{L}_{\text{CLS}} = \frac{1}{N_s} \sum_{i=1}^{N_s} \mathcal{L}_{\text{CE}}(C(\hat{f}_s^i), y_s^i) \tag{8}$$

Since we have no access to the ground-truth labels for the target domain data, it is impossible to use the supervised cross-entropy loss as in Eq. 8 on samples from $\mathcal{T}$ and $\tilde{\mathcal{T}}$. One straightforward alternative is to adopt pseudo-labels as in Fan et al. (2022); Liang et al. (2022); Zou et al. (2021); Shen et al. (2022). However, this strategy has the risk that the classifier might mistakenly predict target samples as the major classes of the source domain. Instead, we use NWD to ensure consistency in predictions between $\mathcal{T}$ and $\mathcal{T}^d$. Thus, we define a target domain loss $\mathcal{L}_{\text{NWD}}^{\mathcal{T}}$ as:

$$\mathcal{L}_{\text{NWD}}^{\mathcal{T}} = \frac{1}{N_t} \sum_{i=1}^{N_t} \|C(f_t^i)\|_* - \frac{1}{N_t} \sum_{i=1}^{N_t} \|C(f_{t^d}^i)\|_* \tag{9}$$

where $f_{t^d}^i \sim \nu_{t^d}$ denotes the deformed target domain batch and $f_t^i \sim \nu_t$ denotes the original target domain batch. Combining Eq. 3, Eq. 8, Eq. 7 and Eq. 9 together, our overall objective loss is:

$$\min_{E, h_{\text{SSL}}, C} \alpha \mathcal{L}_{\text{CLS}} + \gamma \mathcal{L}_{\text{SSL}}$$
$$\min_E \max_C \beta_1 \mathcal{L}_{\text{D-NWD}} + \beta_2 \mathcal{L}_{\text{NWD}}^{\mathcal{T}} \tag{10}$$

where $\alpha, \gamma, \beta_1, \beta_2$ are weighting hyperparameters that can be tuned using the target domain validation set. When computing the overall loss, the additional term $\mathcal{L}_{\text{NWD}}^{\mathcal{T}}$, which corresponds to $W_N(\hat{\nu}_t, \hat{\nu}_{t^d})$, does not influence our theoretical contribution of D-NWD, the bound in Eq. 14. The reason is that Eq. 14 specifically focuses on $W_N(\hat{\nu}_{s\cup s^d}, \hat{\nu}_{t\cup t^d})$, which is empirically represented by $\mathcal{L}_{\text{D-NWD}}$. The term $W_N(\hat{\nu}_t, \hat{\nu}_{t^d})$ (or $\mathcal{L}_{\text{NWD}}^{\mathcal{T}}$) is irrelevant in this context, as its influence is not directly related to the bound's conditions (or the empirical measures) being considered.

# 4 THEORETICAL ANALYSIS

We provide a theoretical justification for our Deformed-based Nuclear-norm Wasserstein Discrepancy (D-NWD). Following Ben-David et al. (2006) and Chen et al. (2022), we perform our analysis in a binary classification scenario, which can be easily adapted to multi-class classification through reduction techniques such as one-vs-all Rifkin & Klautau (2004) or one-vs-one Allwein et al. (2000) approaches. Consider $\{C : \mathbb{R}^n \to [0,1]\}$ as a set of source classifiers within the hypothesis space $\mathcal{H}$. The risk or error of classifier $C$ on the original source domain is defined as $\varepsilon_s(C) = \mathbb{E}_{f_s \sim \nu_s}[|C(f_s) - y_s|]$, where $y_s$ is the label associated with the feature $f_s$. We then define $\varepsilon_{s \cup s^d}(C) = \mathbb{E}_{\hat{f}_s \sim \nu_{s \cup s^d}}[|C(\hat{f}_s) - \hat{y}_s|]$, where $\hat{y}_s$ is the label associated with $\hat{f}_s$. Similarly, we define $\varepsilon_t(C), \varepsilon_{t \cup t^d}(C)$ as the risks on the target domain. The optimal joint hypothesis is defined as $C^* = \arg\min_C \varepsilon_{s \cup s^d}(C) + \varepsilon_t(C)$ which minimizes the combined risk across $\nu_{s \cup s^d}$ and $\nu_t$. Our Theorem 1 demonstrates that the expected target risk $\varepsilon_t(C)$ can be bounded by the D-NWD on $\nu_{s \cup s^d}$ and $\nu_{t \cup t^d}$, $W_N(\nu_{s \cup s^d}, \nu_{t \cup t^d})$. Building on Theorem 1, we derive Theorem 2. Theorem 2 establishes that $\varepsilon_t(C)$ can be bounded by D-NWD on empirical probability measures $\hat{\nu}_{s \cup s^d}$ and $\hat{\nu}_{t \cup t^d}$, $W_N(\hat{\nu}_{s \cup s^d}, \hat{\nu}_{t \cup t^d})$. We prove Lemma 1 and 4 to support our proof of Theorem 1 and 2. **Backgrounds of 1-Wasserstein distance and NWD are in Appendix A. All proofs are included in the Appendix B.**

**Lemma 1.** *Let $(\Omega_1, \mathcal{F}_1, \nu_1)$ and $(\Omega_2, \mathcal{F}_2, \nu_2)$ be two probability spaces, where $\Omega_1, \Omega_2$ are two disjoint sample spaces. Let $p_1, p_2 \in [0,1]$ be constants such that $p_1 + p_2 = 1$. Let $(\Omega_3, \mathcal{F}_3)$ be a measurable space, where $\mathcal{F}_3$ is the $\sigma$-algebra on $\Omega_3 = \Omega_1 \cup \Omega_2$. Then, the measure $\nu_3$ defined on the measurable space $(\Omega_3, \mathcal{F}_3)$ as:*

$$\nu_3(A) = p_1 \nu_1(A \cap \Omega_1) + p_2 \nu_2(A \cap \Omega_2), \quad \forall A \in \mathcal{F}_3 \tag{11}$$

*is a probability measure on $(\Omega_3, \mathcal{F}_3)$.*

**Theorem 1.** *Let $(\Omega_o, \mathcal{F}_o, \nu_s)$, $(\Omega_d, \mathcal{F}_d, \nu_{s^d})$, $(\Omega_o, \mathcal{F}_o, \nu_t)$, and $(\Omega_d, \mathcal{F}_d, \nu_{t^d})$ be four probability spaces, where $\Omega_o$ and $\Omega_d$ are disjoint and $\Omega_o \cup \Omega_d \subseteq \mathbb{R}^n$. With the results of Lemma 1, let $(\Omega_o \cup \Omega_d, \mathcal{F}_u, \nu_{s \cup s^d})$ and $(\Omega_o \cup \Omega_d, \mathcal{F}_u, \nu_{t \cup t^d})$ be two probability spaces with probability measures defined as $\nu_{s \cup s^d} = 1/2\nu_s + 1/2\nu_{s^d}$ and $\nu_{t \cup t^d} = 1/2\nu_t + 1/2\nu_{t^d}$. Specifically, when sampling from $\nu_{t \cup t^d}$, there is an equal probability of $1/2$ to sample from $v_t$ or $v_{t^d}$. Similarly, sampling from $\nu_{s \cup s^d}$ gives an equal probability of $1/2$ to draw from $v_s$ or $v_{s^d}$. Let $K$ denote a Lipschitz constant. Consider a classifier $C \in \mathcal{H}_1$ and an ideal classifier $C^* = \arg\min_C \varepsilon_{s \cup s^d}(C) + \varepsilon_t(C)$ satisfying the $K$-Lipschitz constraint, where $\mathcal{H}_1$ is a subspace of the hypothesis space $\mathcal{H}$. For every classifier $C$ in $\mathcal{H}_1$, the following inequality holds:*

$$\varepsilon_t(C) \leqslant 2\varepsilon_{s \cup s^d}(C) + 4K \cdot W_N(\nu_{s \cup s^d}, \nu_{t \cup t^d}) + \eta^* \tag{12}$$

*where $\eta^* = 2\varepsilon_{s \cup s^d}(C^*) + \varepsilon_t(C^*)$ is the ideal combined risk and is a sufficiently small constant.*

**Definition 3 ($L_1$-Transportation Cost Information Inequality). Djellout et al. (2004)** *Given $\eta > 0$, a probability measure $\nu$ on a measurable space $(\Omega, \mathcal{F})$ satisfies $T_1(\eta)$ if the inequality*

$$W_1(\nu', \nu) \leqslant \sqrt{\frac{2}{\eta} H(\nu'|\nu)} \tag{13}$$

*where $H(\nu'|\nu) = \int \log \frac{d\nu'}{d\nu} d\nu'$ holds for any probability measure $\nu'$ on $(\Omega, \mathcal{F})$, and $W_1$ represents the 1-Wasserstein distance.*

**Lemma 3. (Corollary 2.6 in Bolley & Villani (2005))** *For a probability measure $\nu$ on a measurable space $(\Omega, \mathcal{F})$, the following statements are equivalent:*

- *$\nu$ satisfies $T_1(\eta)$ inequality for some $\eta$ that can be explicitly found.*
- *$\nu$ has a square-exponential moment, i.e., there exists $\alpha > 0$ such that*

$$\int_\Omega \exp(\alpha d(x, y)^2) \, d\nu(x) \text{ is finite}$$

 *for any $y \in \Omega$. Here, $d$ is a measurable distance over $\Omega$.*

**Lemma 4.** *Let $(\Omega_1, \mathcal{F}_1, \nu_1)$ and $(\Omega_2, \mathcal{F}_2, \nu_2)$ be two probability spaces, where $\Omega_1$ and $\Omega_2$ are disjoint. Let $p_1, p_2 \in [0,1]$ be constants such that $p_1 + p_2 = 1$. Let $p_1, p_2 \in [0,1]$ be constants such that $p_1 + p_2 = 1$. Define a new measure $\nu_3$ on a measurable space $(\Omega_3, \mathcal{F}_3)$, where $\Omega_3 = \Omega_1 \cup \Omega_2$:*

$$\nu_3(A) = p_1 \nu_1(A \cap \Omega_1) + p_2 \nu_2(A \cap \Omega_2), \quad \forall A \in \mathcal{F}_3$$

*Suppose that $\nu_1$ and $\nu_2$ each has a square-exponential moment for some $\alpha_1, \alpha_2 > 0$, respectively. Then, $\nu_3$ is a probability measure (according to Lemma 1), and $\nu_3$ has a square-exponential moment for some $0 < \alpha \leqslant \min(\alpha_1, \alpha_2)$.*

**Theorem 2. (Theorem 2 of Redko et al. (2017))** *Under the assumption of Theorem 1, let $(\Omega_o \cup \Omega_d, \mathcal{F}_u, \nu_{s \cup s^d})$ and $(\Omega_o \cup \Omega_d, \mathcal{F}_u, \nu_{t \cup t^d})$ be two probability spaces with $\nu_{s \cup s^d} = 1/2\nu_s + 1/2\nu_{s^d}$ and $\nu_{t \cup t^d} = 1/2\nu_t + 1/2\nu_{t^d}$, where $\nu_s, \nu_{s^d}, \nu_t, \nu_{t^d}$ each has a square-exponential moment. From Lemma 3 and 4, $\nu_{s \cup s^d}$ satsifies $T_1(\eta_s)$ for some $\eta_s$ and $\nu_{t \cup t^d}$ satsifies $T_1(\eta_t)$ for some $\eta_t$. Let $F_s = \{\hat{f}_s^i\}_{i=1}^{N_s}$ and $F_t = \{\hat{f}_t^i\}_{i=1}^{N_t}$ be two sample sets of size $N_s$ and $N_t$ drawn i.i.d from $\nu_{s \cup s^d}$ and $\nu_{t \cup t^d}$, respectively. $\hat{\nu}_{s \cup s^d} = \frac{1}{N_s} \sum_{i=1}^{N_s} \delta_{\hat{f}_s^i}$ and $\hat{\nu}_{t \cup t^d} = \frac{1}{N_t} \sum_{i=1}^{N_t} \delta_{\hat{f}_t^i}$ are associated empirical probability measures. Then, for any $n' > n$ and $\eta' < \min(\eta_s, \eta_t)$, there exists a constant $N_0$ depending on $n'$ such that for any $\delta > 0$ and $\min(N_s, N_t) \geqslant N_0 \max(\delta^{-(n'+2)}, 1)$, with probability at least $1 - \delta$, the following holds for all $C$:*

$$\varepsilon_t(C) \leqslant 2\varepsilon_{s \cup s^d}(C) + 4K \cdot W_N(\hat{\nu}_{s \cup s^d}, \hat{\nu}_{t \cup t^d}) + \eta^* + 4K \cdot \sqrt{\frac{2}{\eta'} \log \frac{1}{\delta}} \left( \sqrt{\frac{1}{N_s}} + \sqrt{\frac{1}{N_t}} \right) \tag{14}$$

*where $\eta^* = 2\varepsilon_{s \cup s^d}(C^*) + \varepsilon_t(C^*)$ is the ideal combined risk and is a sufficiently small constant.*

In Equation 14, $\eta^*$ are sufficiently small constants for relevant domains with consistent labels because $\mathcal{C}^*$ is the error corresponding the ideal classifier. The term $\sqrt{\frac{2}{\eta'} \log \frac{1}{\delta}} \left( \sqrt{\frac{1}{N_s}} + \sqrt{\frac{1}{N_t}} \right)$ is also a small constant when $N_s$ and $N_t$ are large. $\varepsilon_{s \cup s^d}(C)$ is minimized by a supervised classification loss, since source domain samples have labels. Therefore, the primary objective of our UDA task is to minimize our D-NWD on $\hat{\nu}_{s \cup s^d}$ and $\hat{\nu}_{t \cup t^d}$, $W_N(\hat{\nu}_{s \cup s^d}, \hat{\nu}_{t \cup t^d})$. Hence, minimizing D-NWD can improve the model's performance on samples from the original target domain $\mathcal{T}$. Note that we do not claim that our bound (Eq. 14) is tighter than the one in Theorem 2 Chen et al. (2022), which aligns $\hat{\nu}_s$ and $\hat{\nu}_t$, the empirical probability measures of features from samples in $\mathcal{S}$ and $\mathcal{T}$. In fact, a direct comparison between the two bounds is not feasible, as they apply to different probability measures: $(\hat{\nu}_{s \cup s^d}, \hat{\nu}_{t \cup t^d})$ for D-NWD and $(\hat{\nu}_s, \hat{\nu}_t)$ for NWD. Rather than providing a tighter bound, our theoretical contribution lies in the fact that, regardless of the deformation method used, optimizing the D-NWD on $\hat{\nu}_{s \cup s^d}$ and $\hat{\nu}_{t \cup t^d}$ can effectively reduce the error on the samples from $\mathcal{T}$. In other words, D-NWD mitigates the negative effects of domain gaps and enhances performance on $\mathcal{T}$, as NWD does. However, unlike NWD, D-NWD includes features from deformed samples. This inclusion accounts for a more diverse and robust feature space, improving model generalization under domain shifts. Our empirical results show that with carefully designed deformation techniques, like our proposed CurvRec, D-NWD can outperform NWD in practice.

## 5 EXPERIMENTS

We evaluate our method on the **PointDA-10** Qin et al. (2019) dataset, a domain adaptation dataset for point cloud classification, and on **PointSegDA** Achituve et al. (2021), a dataset for point cloud segmentation. For the PointDA-10 dataset, we compare our approach against the recent state-of-the-art methods for point cloud domain adaptation, including **DANN** Ganin et al. (2016), **PointDAN** Qin et al. (2019), **RS** Sauder & Sievers (2019), **DefRec+PCM** Achituve et al. (2021), **GAST** Zou et al. (2021), and **ImplicitPCDA** Shen et al. (2022). Additionally, we incorporate Self-Paced Self-Training (SPST) into GAST, ImplicitPCDA, and our method, as SPST is originally included in both GAST and ImplicitPCDA. For the **PointSegDA** dataset, we compare our method with **RS**, **DefRec+PCM**, **GAST**, **ImplicitPCDA**, and **Adapt-SegMap** Tsai et al. (2018). We exclude SPST for this dataset. The reason is in Appendix D. For both datasets, we also evaluate two upper bounds: **Supervised-T**, which involves training exclusively on labeled target samples, and **Supervised**, which uses both labeled source and target samples. Additionally, we assess a lower bound, **Unsupervised**, which utilizes only labeled source samples.

### 5.1 DATASETS

**PointDA-10** consists of three three domains: ShapeNet-10 Chang et al. (2015), ModelNet-10 Wu et al. (2015), and ScanNet-10 Dai et al. (2017), each sharing ten distinct classes. **PointSegDA** consists of four domains: ADOBE, FAUST, MIT, and SCAPE. These domains share eight distinct classes of human body parts but vary in point distribution, pose, and scanned humans.

## 5.2 TRAINING SCHEME

Following the literature, we use DGCNN as the feature extractor Achituve et al. (2021) for fair comparison. We train the model of each method three times using distinct random seeds for initialization and report the average accuracy and standard deviation. To ensure a fair comparison, we maintain the same seed for data shuffling and use the Adam optimizer Kingma & Ba (2014) for optimization. For hyperparameter settings, please refer to Appendix D for the details.

| Models | MS | MS$^+$ | SM | SS$^+$ | S$^+$M | S$^+$S | Avg |
|---|---|---|---|---|---|---|---|
| Supervised-T | $93.9_{+0.2}$ | $78.4_{+0.6}$ | $96.2_{+0.1}$ | $78.4_{+0.6}$ | $96.2_{+0.4}$ | $93.9_{+0.2}$ | 89.5 |
| Unsupervised | $83.3_{+0.7}$ | $43.8_{+2.3}$ | $75.5_{+1.8}$ | $42.5_{+1.4}$ | $63.8_{+3.9}$ | $64.2_{+0.8}$ | 62.2 |
| DANN | $75.3_{+0.6}$ | $41.5_{+0.2}$ | $62.5_{+1.4}$ | $46.1_{+2.8}$ | $53.3_{+1.2}$ | $63.2_{+1.2}$ | 57.0 |
| PointDAN | $82.5_{+0.8}$ | $47.7_{+1.0}$ | $77.0_{+0.3}$ | $48.5_{+2.1}$ | $55.6_{+0.6}$ | $67.2_{+2.7}$ | 63.1 |
| RS | $81.5_{+1.2}$ | $35.2_{+5.9}$ | $71.9_{+1.4}$ | $39.8_{+0.7}$ | $61.0_{+3.3}$ | $63.6_{+3.4}$ | 58.8 |
| DefRec+PCM | $81.7_{+0.6}$ | $51.8_{+0.3}$ | $78.6_{+0.7}$ | $54.5_{+0.3}$ | $73.7_{+1.6}$ | $71.1_{+1.4}$ | 68.6 |
| GAST | $82.3_{+0.6}$ | $53.0_{+1.1}$ | $72.6_{+1.9}$ | $47.6_{+1.5}$ | $64.6_{+1.5}$ | $66.8_{+0.6}$ | 64.5 |
| GAST+SPST | $84.5_{+0.5}$ | $54.1_{+1.8}$ | $80.1_{+4.6}$ | $46.7_{+0.6}$ | $81.5_{+1.7}$ | $66.7_{+1.1}$ | 68.9 |
| ImplicitPCDA | $79.5_{+0.4}$ | $41.7_{+1.3}$ | $72.9_{+1.0}$ | $47.5_{+2.9}$ | $67.6_{+5.2}$ | $66.4_{+0.9}$ | 62.6 |
| ImplicitPCDA+SPST | $81.3_{+2.2}$ | $33.2_{+13.4}$ | $73.2_{+3.4}$ | $38.0_{+4.6}$ | $66.9_{+7.7}$ | $75.0_{+2.7}$ | 61.3 |
| CDND | $84.1_{+0.3}$ | $\mathbf{58.7}_{+0.8}$ | $76.2_{+0.0}$ | $\mathbf{55.7}_{+1.0}$ | $75.1_{+1.5}$ | $72.0_{+1.9}$ | 70.3 |
| CDND+SPST | $\mathbf{85.4}_{+1.1}$ | $57.6_{+1.3}$ | $\mathbf{85.0}_{+2.2}$ | $54.5_{+1.1}$ | $\mathbf{82.6}_{+0.7}$ | $\mathbf{74.6}_{+4.4}$ | **73.3** |

Table 1: Performance results (accuracy) on PointDA-10 dataset.

| Models | MS | MS$^+$ | SM | SS$^+$ | S$^+$M | S$^+$S | Avg |
|---|---|---|---|---|---|---|---|
| NWD | $83.3_{+0.7}$ | $46.7_{+1.7}$ | $75.5_{+1.8}$ | $48.9_{+2.5}$ | $63.8_{+3.9}$ | $66.7_{+1.9}$ | 64.2 |
| DefRec | $83.4_{+0.5}$ | $46.9_{+2.3}$ | $74.5_{+0.9}$ | $46.3_{+0.6}$ | $67.7_{+2.3}$ | $64.0_{+0.8}$ | 64.0 |
| DefRec+NWD | $83.4_{+0.5}$ | $51.2_{+3.0}$ | $74.5_{+0.9}$ | $53.7_{+3.8}$ | $67.7_{+2.3}$ | $68.5_{+2.4}$ | 66.5 |
| DefRec+D-NWD | $83.4_{+0.5}$ | $53.1_{+2.3}$ | $74.5_{+0.9}$ | $54.6_{+1.0}$ | $67.7_{+2.3}$ | $67.4_{+0.1}$ | 66.8 |
| CurvRec(S)-High | $83.8_{+0.9}$ | $52.0_{+1.4}$ | $\mathbf{78.0}_{+1.0}$ | $45.9_{+3.8}$ | $72.5_{+1.4}$ | $66.7_{+1.1}$ | 66.5 |
| CurvRec(S)-Low | $83.1_{+0.9}$ | $53.0_{+1.9}$ | $74.9_{+0.4}$ | $44.7_{+1.2}$ | $74.8_{+0.9}$ | $65.9_{+0.2}$ | 66.1 |
| CurvRec(En)-High | $82.9_{+1.5}$ | $52.1_{+0.4}$ | $77.0_{+0.3}$ | $46.7_{+1.0}$ | $70.9_{+0.6}$ | $65.8_{+0.4}$ | 65.9 |
| CurvRec(En)-Low | $\mathbf{84.1}_{+0.3}$ | $52.2_{+1.3}$ | $76.2_{+0.0}$ | $50.1_{+0.3}$ | $75.1_{+1.5}$ | $66.4_{+1.5}$ | 67.4 |
| CurvRec(En)-Low+PCM | $83.0_{+0.5}$ | $53.7_{+1.0}$ | $74.0_{+0.6}$ | $54.8_{+1.1}$ | $73.8_{+1.1}$ | $\mathbf{76.8}_{+0.9}$ | 69.4 |
| CurvRec(En)-Low+NWD | $\mathbf{84.1}_{+0.2}$ | $54.3_{+2.2}$ | $76.2_{+0.0}$ | $52.7_{+2.1}$ | $75.1_{+1.5}$ | $70.6_{+2.2}$ | 68.8 |
| CDND (CurvRec(En)-Low+D-NWD) | $84.1_{+0.3}$ | $\mathbf{58.7}_{+0.8}$ | $76.2_{+0.0}$ | $\mathbf{55.7}_{+1.0}$ | $75.1_{+1.5}$ | $72.0_{+1.9}$ | **70.3** |

Table 2: Ablation study results (accuracy) on PointDA-10 dataset.

| Models | FA | FM | FS | MA | MF | MS | AF | AM | AS | SA | SF | SM | AVG |
|---|---|---|---|---|---|---|---|---|---|---|---|---|---|
| Supervised | $80.9_{+7.2}$ | $81.8_{+0.3}$ | $82.4_{+1.2}$ | $80.9_{+7.2}$ | $84.0_{+1.8}$ | $82.4_{+1.2}$ | $84.0_{+1.8}$ | $81.8_{+0.3}$ | $82.4_{+1.2}$ | $80.9_{+7.2}$ | $84.0_{+1.8}$ | $81.8_{+0.3}$ | 82.3 |
| Unsupervised | $78.5_{+0.4}$ | $60.9_{+0.6}$ | $66.5_{+0.6}$ | $26.6_{+3.5}$ | $33.6_{+1.3}$ | $69.9_{+1.2}$ | $38.5_{+2.2}$ | $31.2_{+1.4}$ | $30.0_{+3.6}$ | $74.1_{+1.0}$ | $68.4_{+2.4}$ | $64.5_{+0.5}$ | 53.6 |
| AdaptSegMap | $70.5_{+3.4}$ | $60.1_{+0.6}$ | $65.3_{+1.3}$ | $49.1_{+9.7}$ | $54.0_{+0.5}$ | $62.8_{+7.6}$ | $\mathbf{44.2}_{+1.7}$ | $35.4_{+0.3}$ | $35.1_{+1.4}$ | $70.1_{+2.5}$ | $67.7_{+1.4}$ | $63.8_{+1.2}$ | 56.5 |
| RS | $78.7_{+0.5}$ | $60.7_{+0.4}$ | $\mathbf{66.9}_{+0.4}$ | $59.6_{+5.0}$ | $38.4_{+2.1}$ | $\mathbf{70.4}_{+1.0}$ | $44.0_{+0.6}$ | $30.4_{+0.5}$ | $36.6_{+0.8}$ | $70.7_{+0.8}$ | $\mathbf{73.0}_{+1.5}$ | $\mathbf{65.3}_{+1.3}$ | 57.9 |
| DefRec+PCM | $78.8_{+0.2}$ | $\mathbf{60.9}_{+0.8}$ | $63.6_{+0.1}$ | $48.1_{+0.4}$ | $48.6_{+2.4}$ | $70.1_{+0.8}$ | $46.9_{+1.0}$ | $33.2_{+0.3}$ | $37.6_{+0.1}$ | $66.3_{+1.7}$ | $66.5_{+1.0}$ | $62.6_{+0.2}$ | 56.9 |
| GAST | $76.7_{+2.3}$ | $55.0_{+1.0}$ | $60.3_{+1.0}$ | $52.1_{+4.4}$ | $35.2_{+0.4}$ | $69.6_{+1.2}$ | $43.3_{+3.7}$ | $25.9_{+3.6}$ | $30.8_{+4.0}$ | $57.4_{+10.6}$ | $66.1_{+1.3}$ | $64.6_{+0.5}$ | 53.1 |
| ImplicitPCDA | $47.5_{+0.6}$ | $53.2_{+1.0}$ | $54.2_{+3.4}$ | $51.1_{+1.6}$ | $\mathbf{64.0}_{+1.3}$ | $56.1_{+4.2}$ | $44.1_{+0.9}$ | $\mathbf{42.3}_{+1.3}$ | $\mathbf{40.5}_{+1.2}$ | $49.7_{+2.1}$ | $70.6_{+1.4}$ | $55.0_{+2.5}$ | 52.4 |
| CDND | $\mathbf{81.5}_{+2.0}$ | $60.7_{+0.5}$ | $61.4_{+0.5}$ | $\mathbf{68.6}_{+1.4}$ | $47.2_{+1.4}$ | $67.7_{+1.4}$ | $43.6_{+0.5}$ | $35.3_{+2.2}$ | $40.1_{+1.5}$ | $\mathbf{77.5}_{+0.5s}$ | $70.4_{+1.1}$ | $65.1_{+0.3}$ | **59.9** |

Table 3: Performance results (mIOU) on PointSegDA dataset.

## 5.3 RESULTS

**Results on PointDA.** The results are presented in Table 1. We use S$^+$ to represent the ScanNet dataset, M to represent ModelNet, and S to represent the ShapeNet dataset. The CDND model shows significant improvement over the other approaches on the PointDA-10 dataset with the highest average accuracy of 70.3%, outperforming all other models. CDND delivers state-of-the-art performance on five out of six tasks. It excels in tasks with a large domain gap, such as MS$^+$, S$^+$M, SS$^+$, and S$^+$S. In these tasks, one domain is a synthetic dataset and another domain is a real-world dataset. This shows its proficiency in handling complex transformations. Especially, CDND scores 58.7% on MS$^+$, outperforming the second-best method by approximately 6%. Additionally, CDND

maintains competitive accuracy in tasks with a small domain gap, such as SM and MS, with scores of 84.1% and 76.2%, respectively. With SPST, the performance is further improved, as CDND+SPST achieves 73.3%, outperforming GAST+SPST by 4.4% and ImplicitPCDA+SPST by 12%. Note that plain CDND also outperforms both GAST+SPST and ImplicitPCDA+SPST on average. The strong performance of CDND across various tasks highlights its ability to adapt to diverse domain challenges, making it a promising choice for point cloud classification in the UDA setting.

**Results on PointSegDA.** The results are presented in Table 3 We use A to represent the ADOBE dataset, F to represent the FAUST dataset, M to represent the MIT dataset, and S to represent the SCAPE dataset. On the PointSegDA benchmark, CDND achieves the highest average score of 59.9, which surpasses the second-best method, RS, by a margin of 2.0%, which is significant in terms of mIoU on the segmentation task. Its superior performance is particularly evident in MA and SA tasks; in the MA task, CDND achieves a mIoU of 68.6, outperforming RS by 9%. Similarly, in the SA task, CDND secures a mIoU of 77.5, which is around 7% higher than RS. These results showcase its adaptability and learning capability. Additionally, in the FA task, CDND achieves a score of 81.5, even slightly surpassing the supervised baseline. In other tasks, *i.e.*, FM, AS, and SM tasks, CDND either matches or comes very close to the top-performing models, validating its status as a consistently high-performing model. The widespread dominance across various tasks on the PointSegDA benchmark further emphasizes CDND's effectiveness.

## 5.4 ABLATION STUDY

To demonstrate the effectiveness of each component of CDND, we conduct ablative studies on the PointDA-10 dataset. There are several ways to evaluate curvature diversity. While standard deviation is commonly used to evaluate the diversity of data points, we propose using entropy. We compare our entropy-based approach (CurvRec(En)) with a standard deviation-based method (CurvRec(S)). To validate our hypothesis that focusing on low curvature diversity regions can improve performance, we investigate the impact of deforming areas with both high (CurvRec(En)+High, CurvRec(S)+High) and low (CurvRec(En)+Low, CurvRec(S)+Low) diversity.

**Effectiveness of CurvRec.** From Table 2, when comparing CurvRec(En) variants with CurvRec(S) variants, CurvRec(En) demonstrates better performance, with a more distinct difference between CurvRec(En)-High and CurvRec(En)-Low. This suggests that entropy is a superior method for evaluating curvature diversity in regions. In contrast, there is a much less distinction between CurvRec(S)-High and CurvRec(S)-Low. Notably, all CurvRec measures outperform DefRec, regardless of whether the focus is on high or low curvature diversity. We hypothesize that deforming regions with high curvature diversity can help the model become more robust to changes in these areas, potentially improving performance. However, all "Low" outperforms "High". This suggests that allowing the feature extractor to concentrate on preserving and learning from the semantically rich regions is more beneficial, as altering semantically rich regions can lead to the loss of critical information. Compared to the plain NWD, all CurvRec variants perform better than plain NWD. Specifically, CurvRec(En)-Low surpasses DefRec and NWD by approximately 3%. Though CurvRec(En)-Low demonstrates better performance overall, it does not outperform our proposed CDND. CDND (CurvRec(En)-Low +D-NWD), outperforms all CurvRec variants, DefRec variants, and plain NWD. This highlights the effectiveness of our D-NWD loss. Compared to CurvRec(En)-Low, integrating with D-NWD improves average performance by 2.9%, with specific gains of 6.5% on MS$^+$, 5.6% on SS$^+$, and 5.6% on S$^+$S.

**Effectiveness of D-NWD.** To illustrate the effectiveness of our D-NWD, we compare CDND with two alternatives: CurvRec(En)-Low+PCM, which replaces D-NWD with PCM (PointMixup), and CurvRec(En)-Low+NWD. On average, CDND outperforms both methods. Specifically, compared to CurvRec(En)-Low+PCM, CDND achieves an improvement of approximately 5% on MS$^+$, 1% on SS$^+$, 1% on MS, and 2% on SM. When compared to CurvRec(En)-Low+NWD, CDND surpasses it by 4.4% on MS$^+$, 3% on SS$^+$, and 1.4% on S$^+$S. To further demonstrate that ***our D-NWD can generalize to any deformation method***, we include the results of DefRec+D-NWD and DefRec+NWD. Compared to plain DefRec, DefRec+D-NWD shows an overall improvement of 2.8%, including around 6.2% on MS$^+$, and 3.4% on S$^+$S, and a notable 8.3% on SS$^+$. DefRec+D-NWD also outperforms DefRec+NWD on MS$^+$ and S$^+$S tasks, as well as on average, demonstrating that D-NWD has competitive performance compared to NWD across various deformation methods. Note that D-NWD and NWD effectively improve performance on tasks with large domain gaps, primarily be-

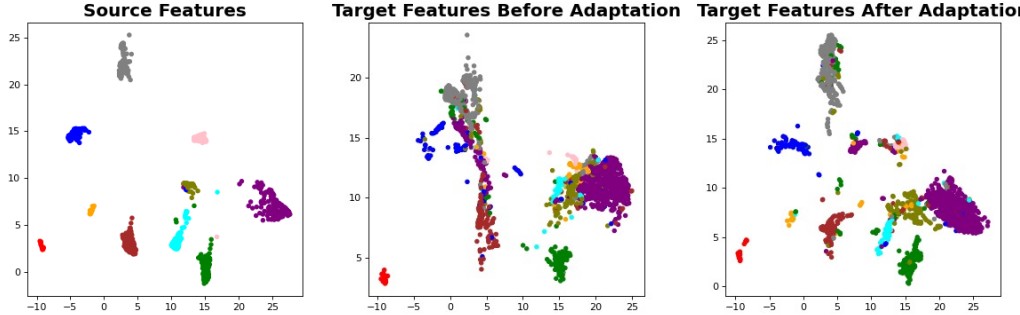

Figure 2: UMAP visualizations depict pre-activation data representations for the MS[+] task, with different colors denoting different classes. The center plot shows the target domain test data representations generated from a model trained on the source dataset without any adaptation. The left and right plots show the source and target domain data representations after adaptation using CDND.

tween real and synthetic dataset pairs such as MS[+] and S[+]S. For tasks like MS, SM, and S[+]M, where the results are already strong with just CurvRec or DefRec, using D-NWD or NWD does not offer significant additional benefits. In fact, plain CurvRec and DefRec perform better in these cases, so we retain the performance of plain CurvRec or DefRec for MS, SM, and S[+]M.

## 5.5 ANALYTIC EXPERIMENTS

We conduct analytical experiments to gain deeper insights into the effectiveness of our approach. Specifically, we assess how CDND impacts the *distribution* of the target domain in the classifier's output space for the challenging ModelNet to ScanNet task (MS[+]); ModelNet is a synthetic dataset, while ScanNet is a real-world dataset, making the domain shift between them particularly challenging. We used UMAP to visualize and compare data representations of validation data from the source domain, and test data from the target domain both before and after applying CDND. Figure 2 shows each point as a data representation in the classifier's output space before softmax activation, with different colors denoting different classes. The middle plot in Figure 2 illustrates that, prior to adaptation, the classifier struggles with the target domain data, as points from different classes are heavily intermixed. However, after applying CDND, the class boundaries become more distinct, and the distribution of target domain representations aligns well with that of the source domain. This improvement is visible in the left and right plots of Figure 2, where the arrangement of points shows a more distinct, consistent pattern across both domains. In other words, we see that the feature space becomes domain-agnostic. This visualization further demonstrates CDND's efficacy in reducing domain shift-induced performance degradation and enhancing class distinction.

## 6 CONCLUSION

We introduced a novel unsupervised domain adaptation approach specifically for point cloud data, which presents unique challenges due to its intricate geometric structures. Our method, CDND, integrates curvature diversity-based deformation with Deformation-based Nuclear-norm Wasserstein discrepancy (D-NWD) to mitigate target domain performance degradation. Our theoretical analysis of D-NWD shows it minimizes an upper bound for target domain model error, thus enhancing performance. Additionally, the theoretical analysis shows that D-NWD can be applied to ***any*** deformation method. Experimental results indicate that our approach is highly effective, surpassing state-of-the-art methods on two major benchmarks. The success of our method in handling large domain differences highlights its adaptability and robustness. Ablation studies confirm that both core components of CDND are essential for achieving optimal performance. Future work could explore extending our approach to scenarios where both source domain data are not directly accessible due to privacy or when the two domains share a subset of their classes.

## REPRODUCIBILITY STATEMENT

For reproducibility, detailed implementation specifications are available in the Appendix. we have included our source codes and environment setup instructions in the supplementary materials.

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

## A   REVIEW OF THE 1-WASSERSTEIN DISTANCE AND NWD

We begin the review of Nuclear-norm Wasserstein by introducing 1-Wasserstein distance:

**Definition 1 (1-Wasserstein distance). Adler & Lunz (2018)** *1-Wasserstein distance quantifies the minimal cost of transporting mass between two probability measures that are defined on the same sample space. Let $\mu$ and $\nu$ be two probability measures over $\Omega$. Let $(\Omega, d)$ be a metric space, where $d(x, y)$ is distance between two points $x$ and $y$ in $\Omega$. $W_1(\mu, \nu)$ is formally defined as:*

$$W_1(\mu, \nu) = \inf_{\gamma \in \Gamma(\mu, \nu)} \int_{\Omega \times \Omega} d(x, y) \, d\gamma(x, y)$$

*where $\Gamma(\mu, \nu)$ is the set of all couplings of $\mu$ and $\nu$. A coupling $\gamma \in \Gamma(\mu, \nu)$ is a joint probability distribution on $\Omega \times \Omega$ with marginals $\mu$ and $\nu$, meaning:*

$$\int_A \int_\Omega \gamma(x, y) \, dydx = \mu(A) \quad and \quad \int_A \int_\Omega \gamma(x, y) \, dxdy = \nu(A), \quad \forall A \in \mathcal{F}$$

*Kantorovich-Rubinstein Duality shows that $W_1(\mu, \nu)$ can be rewritten as:*

$$W_1(\mu, \nu) = \sup_{\|h\|_L \leqslant K} \mathbb{E}_{x \sim \mu}[h(x)] - \mathbb{E}_{x \sim \nu}[h(x)] \tag{15}$$

*where $\| \cdot \|_L$ is the Lipschitz norm and $K$ is the Lipschitz constant.*

The Nuclear-norm Wasserstein Discrepancy (NWD) Chen et al. (2022) belongs to the family of 1-Wasserstein distances, with a sophisticatedly chosen $h$. Below, we present the form of $h$ in NWD. Consider a prediction matrix $P \in \mathbb{R}^{b \times K}$ predicted by the classifier $C$, where $b$ represents the number of samples in a batch and $K$ represents the number of classes. The non-negative self-correlation matrix $Z \in \mathbb{R}^{K \times K}$ is computed as $Z = P^T P$. The intra-class correlation $I_a$ is defined as the sum of the main diagonal elements of $Z$, and the inter-class correlation $I_e$ is the sum of all the off-diagonal elements of $Z$:

$$I_a = \sum_{i=1}^{K} Z_{ii}, \quad I_e = \sum_{i \neq j}^{K} Z_{ij}$$

In the source domain, $I_a$ is large, and $I_e$ is relatively small because most samples are correctly classified. Conversely, in the target domain, $I_a$ is small, and $I_e$ is relatively large due to the lack of supervised training on the target domain. Hence, $I_a - I_e$ can represent the discrepancy between the two domains, as $I_a - I_e$ is large for the source domain but small for the target domain. Note that $I_a = \|P\|_F^2$ can be represented as the squared Frobenius norm of $P$, and thus $I_a - I_e = \|P\|_F^2 - b$. [2] We can rewrite $P_s = C(f_s)$ and $P_t = C(f_t)$, where $f_s$ and $f_t$ are feature representation batches from the source and target domains, respectively. From the above analysis, we find $\|C\|_F$ gives high scores to the source domain and low scores to the target domain, so $\|C\|_F$ works as a critic function. Thus, we can set $h$ in Eq. 15 to be $\|C\|_F$ and represent the domain discrepancy as:

$$W_F(\nu_s, \nu_t) = \sup_{\|\|C\|_F\|_L \leqslant K} \mathbb{E}_{f_s \sim \nu_s}[\|C(f_s)\|_F] - \mathbb{E}_{f_t \sim \nu_t}[\|C(f_t)\|_F]$$

where $\nu_s$ is the probability measure for features of samples in $\mathcal{S}$ and $\nu_t$ is the probability measure for features of samples in $\mathcal{T}$. To enhance prediction diversity, the Frobenius norm can be replaced with the nuclear norm which maximizes the rank of $P$ while still being bounded by the Frobenius norm Chen et al. (2022). Thus, the domain discrepancy can be rewritten as:

$$W_N(\nu_s, \nu_t) = \sup_{\|\|C\|_* \|_L \leqslant K} \mathbb{E}_{f_s \sim \nu_s}[\|C(f_s)\|_*] - \mathbb{E}_{f_t \sim \nu_t}[\|C(f_t)\|_*] \tag{16}$$

The Eq. 16 is the formal definition of NWD. It can be approximate by $\mathcal{L}_{\text{NWD}}$:

$$\mathcal{L}_{\text{NWD}} = \frac{1}{N_s} \sum_{i=1}^{N_s} \|C(f_s^i)\|_* - \frac{1}{N_t} \sum_{i=1}^{N_t} \|C(f_t^i)\|_*$$

where $f_s^i \sim \nu_s$ represents the features for the $i$-th source batch and $f_t^i \sim \nu_t$ represents the features for the $i$-th target batch.

$$\min_E \max_C \mathcal{L}_{\text{NWD}} \tag{17}$$

Then, the distribution alignment is achieved through a min-max game presented in Eq. 17.

---

[2]We have $\sum_{j=1}^{K} Z_{i,j} = 1, \forall i \in \{1, \cdots, b\}$ and $j \in \{1, \cdots, K\}$, and thus $I_a + I_e = b$ Chen et al. (2022).

# B  THEORETICAL ANALYSIS: PROOFS FOR THEOREMS

In this section, we first prove Theorem 1, which serves as the foundation for Theorem 2. Our proofs are structured as follows: we begin by proving Lemma 1, which supports a key assumption in Theorem 1. Next, we present the proof of Theorem 1. After proving Theorem 1, we prove Lemma 4 and conclude with the proof of Theorem 2.

**Definition 2 (Probability Spacce). Durrett (2019)** *A probability space is a triple $(\Omega, \mathcal{F}, \nu)$. $\Omega$ represents the sample space, the set of all possible outcomes. $\mathcal{F}$ represents the set of events and is a $\sigma$-algebra, which is a collection of all subsets of $\Omega$. $\mathcal{F}$ is closed under complements and countable unions. $\nu$ represents a probability measure on the measurable space $(\Omega, \mathcal{F})$. It is a function $\nu : \mathcal{F} \to [0, 1]$ that assigns to each event $A \in \mathcal{F}$ a real value $\nu(A)$ (the probability of A). $\nu$ satisfies the following three axioms:*

- *Non-negativity: For every event $A \in \mathcal{F}$, $\nu(A) \geqslant v(\varnothing) = 0$*
- *Normalization: $\nu(\Omega) = 1$*
- *$\sigma$-additivity (Countable Additivity): For any countable sequence of pairwise disjoint events $A_1, A_2, A_3, \cdots \in \mathcal{F}$ (where $A_i \cap A_j = \varnothing$ for $i \neq j$),*

$$\nu \left( \bigcup_{i=1}^{\infty} A_i \right) = \sum_{i=1}^{\infty} \nu(A_i)$$

**Lemma 1.** *Let $(\Omega_1, \mathcal{F}_1, \nu_1)$ and $(\Omega_2, \mathcal{F}_2, \nu_2)$ be two probability spaces, where $\Omega_1, \Omega_2$ are two disjoint sample spaces. Let $p_1, p_2 \in [0, 1]$ be constants such that $p_1 + p_2 = 1$. Let $(\Omega_3, \mathcal{F}_3)$ be a measurable space, where $\mathcal{F}_3$ is the $\sigma$-algebra on $\Omega_3 = \Omega_1 \cup \Omega_2$. Then, the measure $\nu_3$ defined on the measurable space $(\Omega_3, \mathcal{F}_3)$ as:*

$$\nu_3(A) = p_1 \nu_1(A \cap \Omega_1) + p_2 \nu_2(A \cap \Omega_2), \quad \forall A \in \mathcal{F}_3$$

*is a probability measure on $(\Omega_3, \mathcal{F}_3)$.*

*Proof.* Since $\nu_1$ and $\nu_2$ are probability measures, they satisfy $\nu_1(B) \geqslant 0$ for all $B \in \mathcal{F}_1$ and $\nu_2(C) \geqslant 0$ for all $C \in \mathcal{F}_2$. For any set $A \in \mathcal{F}_3$, we have:

$$\nu_3(A) = p_1 \nu_1(A \cap \Omega_1) + p_2 \nu_2(A \cap \Omega_2)$$

Given that $p_1, p_2 \geqslant 0$ and $\nu_1(A \cap \Omega_1) \geqslant 0$ and $\nu_2(A \cap \Omega_2) \geqslant 0$, it follows that $\nu(A) \geqslant 0$. Thus, $\nu$ is non-negative. Then, we need to show that $\nu(\Omega_1 \cup \Omega_2) = 1$. Consider:

$$\nu_3(\Omega_1 \cup \Omega_2) = p_1 \nu_1((\Omega_1 \cup \Omega_2) \cap \Omega_1) + p_2 \nu_2((\Omega_1 \cup \Omega_2) \cap \Omega_2)$$

Since $(\Omega_1 \cup \Omega_2) \cap \Omega_1 = \Omega_1$ and $(\Omega_1 \cup \Omega_2) \cap \Omega_2 = \Omega_2$, and $\nu_1(\Omega_1) = 1$ and $\nu_2(\Omega_2) = 1$, we have:

$$\nu_3(\Omega_1 \cup \Omega_2) = p_1 \cdot 1 + p_2 \cdot 1 = p_1 + p_2 = 1$$

Thus, $\nu_3$ is normalized. Let $\{A_i\}_{i=1}^{\infty}$ be a countable collection of pairwise disjoint sets in $\mathcal{F}_3$. The last part is to show $\sigma$-additivity:

$$\nu_3 \left( \bigcup_{i=1}^{\infty} A_i \right) = \sum_{i=1}^{\infty} \nu_3(A_i)$$

By definition of $\nu_3$,

$$\nu_3 \left( \bigcup_{i=1}^{\infty} A_i \right) = p_1 \nu_1 \left( \left( \bigcup_{i=1}^{\infty} A_i \right) \cap \Omega_1 \right) + p_2 \nu_2 \left( \left( \bigcup_{i=1}^{\infty} A_i \right) \cap \Omega_2 \right)$$

Since the $A_i$ are pairwise disjoint, $(\bigcup_{i=1}^{\infty} A_i) \cap \Omega_1 = \bigcup_{i=1}^{\infty} (A_i \cap \Omega_1)$, and similarly for $\Omega_2$. Using the $\sigma$-additivity of $\nu_1$ and $\nu_2$:

$$p_1 \nu_1 \left( \bigcup_{i=1}^{\infty} (A_i \cap \Omega_1) \right) = p_1 \sum_{i=1}^{\infty} \nu_1(A_i \cap \Omega_1)$$

$$p_2 \nu_2 \left( \bigcup_{i=1}^{\infty} (A_i \cap \Omega_2) \right) = p_2 \sum_{i=1}^{\infty} \nu_2(A_i \cap \Omega_2)$$

Thus,

$$\nu_3 \left( \bigcup_{i=1}^{\infty} A_i \right) = p_1 \sum_{i=1}^{\infty} \nu_1(A_i \cap \Omega_1) + p_2 \sum_{i=1}^{\infty} \nu_2(A_i \cap \Omega_2)$$

$$= \sum_{i=1}^{\infty} \left( p_1 \nu_1(A_i \cap \Omega_1) + p_2 \nu_2(A_i \cap \Omega_2) \right)$$

Since $\nu_3(A_i) = p_1 \nu_1(A_i \cap \Omega_1) + p_2 \nu_2(A_i \cap \Omega_2)$, we get:

$$\nu_3 \left( \bigcup_{i=1}^{\infty} A_i \right) = \sum_{i=1}^{\infty} \nu_3(A_i)$$

Thus, $\nu_3$ satisfies $\sigma$-additivity. Since $\nu_3$ satisfies non-negativity, normalization, and $\sigma$-additivity, by definition, $\nu_3$ is a valid probability measure.

**Lemma 2 (Lemma 1 Chen et al. (2022)).** *Let $\nu, \nu'$ be two probability measures on $(\Omega, \mathcal{F})$. Let $d(x, y)$ be the distance between $x \sim \nu$ and $y \sim \nu'$. $W_N$ represents the NWD, and $K$ denotes a Lipschitz constant. Given a family of classifiers $C \in \mathcal{H}_1$ and a ideal classifier $C^* \in \mathcal{H}_1$ satisfying the $K$-Lipschitz constraint, where $\mathcal{H}_1$ is a subspace of $\mathcal{H}$, the following holds for every $C, C^* \in \mathcal{H}_1$.*

$$|\varepsilon(C, C^*) - \varepsilon'(C, C^*)| \leqslant 2K \cdot W_N(\nu_1, \nu_2)$$

*where $\varepsilon(C, C^*) = \mathbb{E}_{x \sim \nu}[|C(x) - C^*(x)|]$ and $\varepsilon'(C, C^*) = \mathbb{E}_{y \sim \nu'}[|C(y) - C^*(y)|]]$.*

For future notations, we define the following:

$$\varepsilon_s(C_1, C_2) = \mathbb{E}_{f_s \sim \nu_s} \left[ |C_1(f_s) - C_2(f_s)| \right]$$

$$\varepsilon_{s \cup s^d}(C_1, C_2) = \mathbb{E}_{\hat{f}_s \sim \nu_{s \cup s^d}} \left[ |C_1(\hat{f}_s) - C_2(\hat{f}_s)| \right]$$

where $C_1, C_2$ are two classifiers in $\mathcal{H}_1$, We define $\varepsilon_t(C_1, C_2)$ and $\varepsilon_{t \cup t^d}(C_1, C_2)$ in the same manner.

**Theorem 1.** *Let $(\Omega_o, \mathcal{F}_o, \nu_s)$, $(\Omega_d, \mathcal{F}_d, \nu_{s^d})$, $(\Omega_o, \mathcal{F}_o, \nu_t)$, and $(\Omega_d, \mathcal{F}_d, \nu_{t^d})$ be four probability spaces, where $\Omega_o$ and $\Omega_d$ are disjoint and $\Omega_o \cup \Omega_d \subseteq \mathbb{R}^n$. With the results of Lemma 1, let $(\Omega_o \cup \Omega_d, \mathcal{F}_u, \nu_{s \cup s^d})$ and $(\Omega_o \cup \Omega_d, \mathcal{F}_u, \nu_{t \cup t^d})$ be two probability spaces with probability measures defined as $\nu_{s \cup s^d} = 1/2 \nu_s + 1/2 \nu_{s^d}$ and $\nu_{t \cup t^d} = 1/2 \nu_t + 1/2 \nu_{t^d}$. Specifically, when sampling from $\nu_{t \cup t^d}$, there is an equal probability of $1/2$ to sample from $v_t$ or $v_{t^d}$. Similarly, sampling from $\nu_{s \cup s^d}$ gives an equal probability of $1/2$ to draw from $v_s$ or $v_{s^d}$. Let $K$ denote a Lipschitz constant. Consider a classifier $C \in \mathcal{H}_1$ and an ideal classifier $C^* = \arg\min_C \varepsilon_{s \cup s^d}(C) + \varepsilon_t(C)$ satisfying the $K$-Lipschitz constraint, where $\mathcal{H}_1$ is a subspace of the hypothesis space $\mathcal{H}$. For every classifier $C$ in $\mathcal{H}_1$, the following inequality holds:*

$$\varepsilon_t(C) \leqslant 2\varepsilon_{s \cup s^d}(C) + 4K \cdot W_N(\nu_{s \cup s^d}, \nu_{t \cup t^d}) + \eta^*$$

*where $\eta^* = 2\varepsilon_{s \cup s^d}(C^*) + \varepsilon_t(C^*)$ is the ideal combined risk and is a sufficiently small constant.*

*Proof.* Let $Z$ be an indicator random variable that indicates whether the sample $\hat{f}_t$ is drawn from $\nu_t$ or $\nu_{t^d}$:

- $Z = 0$ if the sample is from $\nu_{t^d}$.
- $Z = 1$ if the sample is from $\nu_t$.

By the Law of Total Expectation, we have:

$$\varepsilon_{t \cup t^d}(C, C^*) = \mathbb{E}_{\hat{f}_t \sim \nu_{t \cup t^d}}[|C(\hat{f}_t) - C^*(\hat{f}_t)|]$$

$$= \mathbb{E}_{\hat{f}_t \sim \nu_{t \cup t^d}}[|C(\hat{f}_t) - C^*(\hat{f}_t)| \mid Z = 0]P(Z = 0)$$

$$+ \mathbb{E}_{\hat{f}_t \sim \nu_{t \cup t^d}}[|C(\hat{f}_t) - C^*(\hat{f}_t)| \mid Z = 1]P(Z = 1)$$

Substituting $P(Z = 0) = p_0$ and $P(Z = 1) = p_1$,

$$\varepsilon_{t \cup t^d}(C, C^*) = p_0 \mathbb{E}_{\hat{f}_t \sim \nu_{t \cup t^d}}[|C(\hat{f}_t) - C^*(\hat{f}_t)| \mid Z = 0]$$
$$+ p_1 \mathbb{E}_{\hat{f}_t \sim \nu_{t \cup t^d}}[|C(\hat{f}_t) - C^*(\hat{f}_t)| \mid Z = 1]$$

Recognize that $\mathbb{E}_{\hat{f}_t \sim \nu_{t \cup t^d}}[|C(\hat{f}_t) - C^*(\hat{f}_t)| \mid Z = 1]$ is the expectation when $\hat{f}_t$ is drawn from $\nu_t$,

$$\varepsilon_t(C, C^*) = \mathbb{E}_{\hat{f}_t \sim \nu_{t \cup t^d}}[|C(\hat{f}_t) - C^*(\hat{f}_t)| \mid Z = 1]$$

Combining these, we get:

$$\varepsilon_{t \cup t^d}(C, C^*) = p_0 \mathbb{E}_{\hat{f}_t \sim \nu_{t \cup t^d}}[|C(\hat{f}_t) - C^*(\hat{f}_t)| \mid Z = 0] + p_1 \varepsilon_t(C, C^*)$$
$$\frac{1}{p_1} \varepsilon_{t \cup t^d}(C, C^*) = \frac{p_0}{p_1} \mathbb{E}_{\hat{f}_t \sim \nu_{t \cup t^d}}[|C(\hat{f}_t) - C^*(\hat{f}_t)| \mid Z = 0] + \varepsilon_t(C, C^*)$$

Since $\frac{p_0}{p_1} \mathbb{E}_{\hat{f}_t \sim \tilde{\mathcal{T}} \cup \tilde{\mathcal{T}}^d}[|C(\hat{f}_t) - C^*(\hat{f}_t)| \mid Z = 0] \geqslant 0$,

$$\frac{1}{p_1} \varepsilon_{t \cup t^d}(C, C^*) \geqslant \varepsilon_t(C, C^*)$$

Substituting $p_1 = 1/2$, we obtain:

$$2\varepsilon_{t \cup t^d}(C, C^*) \geqslant \varepsilon_t(C, C^*)$$

Based on Lemma 2, we have:

$$|\varepsilon_{s \cup s^d}(C, C^*) - \varepsilon_{t \cup t^d}(C, C^*)| \leqslant 2K \cdot W_N(\nu_{s \cup s^d}, \nu_{t \cup t^d})$$

By triangular inequality,

$$\varepsilon_t(C) \leqslant \varepsilon_t(C^*) + \varepsilon_t(C^*, C)$$
$$\varepsilon_{s \cup s^d}(C, C^*) \leqslant \varepsilon_{s \cup s^d}(C) + \varepsilon_{s \cup s^d}(C^*)$$

Then, we can derive:

$$\begin{aligned}
\varepsilon_t(C) &\leqslant \varepsilon_t(C^*) + \varepsilon_t(C^*, C) \\
&\leqslant \varepsilon_t(C^*) + 2\varepsilon_{t \cup t^d}(C^*, C) \\
&= \varepsilon_t(C^*) + 2\varepsilon_{s \cup s^d}(C, C^*) + 2\varepsilon_{t \cup t^d}(C, C^*) - 2\varepsilon_{s \cup s^d}(C, C^*) \\
&\leqslant \varepsilon_t(C^*) + 2\varepsilon_{s \cup s^d}(C, C^*) + 4K \cdot W_N(\nu_{s \cup s^d}, \nu_{t \cup t^d}) \\
&\leqslant \varepsilon_t(C^*) + 2\varepsilon_{s \cup s^d}(C) + 2\varepsilon_{s \cup s^d}(C^*) + 4K \cdot W_N(\nu_{s \cup s^d}, \nu_{t \cup t^d}) \\
&= 2\varepsilon_{s \cup s^d}(C) + 4K \cdot W_N(\nu_{s \cup s^d}, \nu_{t \cup t^d}) + \eta^*
\end{aligned}$$

**Definition 3 ($L_1$-Transportation Cost Information Inequality). Djellout et al. (2004)** *Given $\eta > 0$, a probability measure $\nu$ on a measurable space $(\Omega, \mathcal{F})$ satisfies $T_1(\eta)$ if the inequality*

$$W_1(\nu', \nu) \leqslant \sqrt{\frac{2}{\eta} H(\nu' | \nu)}$$

*where*

$$H(\nu' | \nu) = \int \log \frac{d\nu'}{d\nu} d\nu'$$

*holds for any probability measure $\nu'$ on $(\Omega, \mathcal{F})$, where $W_1$ represents the 1-Wasserstein distance.*

**Lemma 3. (Corollary 2.6 in Bolley & Villani (2005))** *For a probability measure $\nu$ on a measurable space $(\Omega, \mathcal{F})$, the following statements are equivalent:*

- *$\nu$ satisfies $T_1(\eta)$ inequality for some $\eta$ that can be explicitly found.*
- *$\nu$ has a square-exponential moment, i.e., there exists $\alpha > 0$ such that*

$$\int_{\Omega} \exp(\alpha d(x, y)^2) \, d\nu(x) \text{ is finite}$$

*for any $y \in \Omega$. Here, $d$ is a measurable distance over $\Omega$.*

**Lemma 4.** *Let $(\Omega_1, \mathcal{F}_1, \nu_1)$ and $(\Omega_2, \mathcal{F}_2, \nu_2)$ be two probability spaces, where $\Omega_1$ and $\Omega_2$ are disjoint. Let $p_1, p_2 \in [0,1]$ be constants such that $p_1 + p_2 = 1$. Define a new measure $\nu_3$ on a measurable space $(\Omega_3, \mathcal{F}_3)$, where $\Omega_3 = \Omega_1 \cup \Omega_2$:*

$$\nu_3(A) = p_1 \nu_1(A \cap \Omega_1) + p_2 \nu_2(A \cap \Omega_2), \quad \forall A \in \mathcal{F}_3$$

*Suppose that $\nu_1$ and $\nu_2$ each has a square-exponential moment:*

$$\int_{\Omega_1} \exp(\alpha_1 d_1(x, y_1)^2)\, d\nu_1(x) < \infty, \quad \forall y_1 \in \Omega_1$$

$$\int_{\Omega_2} \exp(\alpha_2 d_2(x, y_2)^2)\, d\nu_2(x) < \infty, \quad \forall y_2 \in \Omega_2$$

*for some $\alpha_1, \alpha_2 > 0$, where $d_1$ is defined over $\Omega_1$ and $d_2$ is defined over $\Omega_2$. Then, $\nu_3$ is a probability measure (according to Lemma 1), and $\nu_3$ has a square-exponential moment for some $0 < \alpha \leqslant \min(\alpha_1, \alpha_2)$.*

*Proof.* First, we define $d : \Omega_3 \times \Omega_3 \to \mathbb{R}^+$:

$$d(x, y) = \begin{cases} d_1(x, y) & \text{if } x, y \in \Omega_1 \\ d_2(x, y) & \text{if } x, y \in \Omega_2 \\ C & \text{if } x \in \Omega_1 \text{ and } y \in \Omega_2 \text{ (or vice versa)} \end{cases}$$

where $C$ is a finite constant chosen to ensure $d$ is a metric on $\Omega_3$. $d$ can be expressed as:

$$d(x, y) = d_1(x, y)\mathbf{1}_{x, y \in \Omega_1} + d_2(x, y)\mathbf{1}_{x, y \in \Omega_2} + C\mathbf{1}_{x \in \Omega_1, y \in \Omega_2 \text{ or } x \in \Omega_2, y \in \Omega_1}$$

where $\mathbf{1}$ is the indicator function. $d_1$ and $d_2$ are measurable by assumption. The indicator functions are measurable because $\Omega_1 \times \Omega_1$, $\Omega_2 \times \Omega_2$, and $(\Omega_1 \times \Omega_2) \cup (\Omega_2 \times \Omega_1)$ are all in the product $\sigma$-algebra $\mathcal{F}_3 \otimes \mathcal{F}_3$. Therefore, $d$ is a sum of products of measurable functions. Hence, $d$ is measurable. Now, for any $y$ in $\Omega_1$,

$$\int_{\Omega_3} \exp(\alpha d(x, y)^2)\, d\nu_3(x)$$

$$= p_1 \int_{\Omega_1} \exp(\alpha d_1(x, y)^2)\, d\nu_1(x) + p_2 \int_{\Omega_2} \exp(\alpha d(x, y)^2)\, d\nu_2(x)$$

$$\leqslant p_1 \int_{\Omega_1} \exp(\alpha_1 d_1(x, y)^2)\, d\nu_1(x) + p_2 \int_{\Omega_2} \exp(\alpha C^2)\, d\nu_2(x)$$

$$= p_1 \int_{\Omega_1} \exp(\alpha_1 d_1(x, y)^2)\, d\nu_1(x) + p_2 \exp(\alpha C^2) < \infty$$

For any $y$ in $\Omega_2$,

$$\int_{\Omega_3} \exp(\alpha d(x, y)^2)\, d\nu_3(x)$$

$$= p_1 \int_{\Omega_1} \exp(\alpha d(x, y)^2)\, d\nu_1(x) + p_2 \int_{\Omega_2} \exp(\alpha d_2(x, y)^2)\, d\nu_2(x)$$

$$\leqslant p_1 \int_{\Omega_1} \exp(\alpha C^2)\, d\nu_1(x) + p_2 \int_{\Omega_2} \exp(\alpha_2 d_2(x, y)^2)\, d\nu_2(x)$$

$$= p_1 \exp(\alpha C^2) + p_2 \int_{\Omega_2} \exp(\alpha_2 d_2(x, y)^2)\, d\nu_2(x) < \infty$$

This proves that $\nu_3$ has a square-exponential moment for some $0 < \alpha \leqslant \min(\alpha_1, \alpha_2)$.

**Lemma 5. (Theorem 1.1 of Bolley et al. (2007); Theorem 1 of Redko et al. (2017))** *Let $\nu$ be a probability measure on $(\Omega, \mathcal{F})$ where $\Omega \subseteq \mathbb{R}^n$. $\nu$ satisfies a $T_1(\eta)$ inequality. Let $\hat{\nu} = \frac{1}{N} \sum_{i=1}^{N} \delta_{f^i}$ be its associated empirical measure defined on a sample set $\{f^i\}_{i=1}^{N}$ of size $N$ drawn i.i.d from $\nu$. Then for any $n' > n$ and $\eta' < \eta$, there exists some constant $N_0$ depending on $n'$*

*and some square-exponential moment of $\nu$ such that for any $\epsilon > 0$ and $N \geqslant N_0 \max(\epsilon^{-(n'+2)}, 1)$, the following holds:*

$$\mathbb{P}[W_N(\nu, \hat{\nu}) > \epsilon] \leqslant \exp\left(-\frac{\eta'}{2} N \epsilon^2\right)$$

**Theorem 2. (Theorem 2 of Redko et al. (2017))** *Under the assumption of Theorem 1, let $(\Omega_o \cup \Omega_d, \mathcal{F}_u, \nu_{s \cup s^d})$ and $(\Omega_o \cup \Omega_d, \mathcal{F}_u, \nu_{t \cup t^d})$ be two probability spaces with $\nu_{s \cup s^d} = 1/2\nu_s + 1/2\nu_{s^d}$ and $\nu_{t \cup t^d} = 1/2\nu_t + 1/2\nu_{t^d}$, where $\nu_s, \nu_{s^d}, \nu_t, \nu_{t^d}$ each has a square-exponential moment. From Lemma 3 and 4, $\nu_{s \cup s^d}$ satisfies $T_1(\eta_s)$ for some $\eta_s$ and $\nu_{t \cup t^d}$ satisfies $T_1(\eta_t)$ for some $\eta_t$. Let $F_s = \{\hat{f}_s^i\}_{i=1}^{N_s}$ and $F_t = \{\hat{f}_t^i\}_{i=1}^{N_t}$ be two sample sets of size $N_s$ and $N_t$ drawn i.i.d from $\nu_{s \cup s^d}$ and $\nu_{t \cup t^d}$, respectively. $\hat{\nu}_{s \cup s^d} = \frac{1}{N_s} \sum_{i=1}^{N_s} \delta_{\hat{f}_s^i}$ and $\hat{\nu}_{t \cup t^d} = \frac{1}{N_t} \sum_{i=1}^{N_t} \delta_{\hat{f}_t^i}$ are associated empirical probability measures. Then, for any $n' > n$ and $\eta' < \min(\eta_s, \eta_t)$, there exists a constant $N_0$ depending on $n'$ such that for any $\delta > 0$ and $\min(N_s, N_t) \geqslant N_0 \max(\delta^{-(n'+2)}, 1)$, with probability at least $1 - \delta$, the following holds for all $C$:*

$$\varepsilon_t(C) \leqslant 2\varepsilon_{s \cup s^d}(C) + 4K \cdot W_N(\hat{\nu}_{s \cup s^d}, \hat{\nu}_{t \cup t^d}) + \eta^* + 4K \cdot \sqrt{\frac{2}{\eta'} \log \frac{1}{\delta}} \left(\sqrt{\frac{1}{N_s}} + \sqrt{\frac{1}{N_t}}\right)$$

*where $\eta^* = 2\varepsilon_{s \cup s^d}(C^*) + \varepsilon_t(C^*)$ is the ideal combined risk and is a sufficiently small constant.*

*Proof.* Based on Theorem 1,

$$\varepsilon_t(C) \leqslant 2\varepsilon_{s \cup s^d}(C) + 4K \cdot W_N(\nu_{s \cup s^d}, \nu_{t \cup t^d}) + \eta^*$$

As a part of a broader class of Wasserstein distances, $W_N$ satisfies the axioms of a distance Villani et al. (2009). Hence, $W_N$ satisfies the triangle inequality:

$$\varepsilon_t(C) \leqslant 2\varepsilon_{s \cup s^d}(C) + 4K \cdot W_N(\nu_{s \cup s^d}, \hat{\nu}_{s \cup s^d}) + 4K \cdot W_N(\hat{\nu}_{s \cup s^d}, \nu_{t \cup t^d}) + \eta^*$$
$$\leqslant 2\varepsilon_{s \cup s^d}(C) + 4K \cdot W_N(\nu_{s \cup s^d}, \hat{\nu}_{s \cup s^d}) + 4K \cdot W_N(\hat{\nu}_{s \cup s^d}, \hat{\nu}_{t \cup t^d})$$
$$+ 4K \cdot W_N(\hat{\nu}_{t \cup t^d}, \nu_{t \cup t^d}) + \eta^*$$

According to Theorem 1, $\Omega_o \cup \Omega_d \subseteq \mathbb{R}^n$. Thus, from Lemma 5,

$$W_N(\nu_{s \cup s^d}, \hat{\nu}_{s \cup s^d}) \leqslant \sqrt{\frac{2}{\eta'} \log\left(\frac{1}{\delta}\right)} \cdot \sqrt{\frac{1}{N_s}}$$

$$W_N(\nu_{t \cup t^d}, \hat{\nu}_{t \cup t^d}) \leqslant \sqrt{\frac{2}{\eta'} \log\left(\frac{1}{\delta}\right)} \cdot \sqrt{\frac{1}{N_t}}$$

$W_N$ belongs to the family of 1-Wasserstein distance. By the symmetry property of distance,

$$W_N(\hat{\nu}_{t \cup t^d}, \nu_{t \cup t^d}) = W_N(\nu_{t \cup t^d}, \hat{\nu}_{t \cup t^d}) \leqslant \sqrt{\frac{2}{\eta'} \log\left(\frac{1}{\delta}\right)} \cdot \sqrt{\frac{1}{N_t}}$$

Substituting back, we have:

$$\varepsilon_t(C) \leqslant 2\varepsilon_{s \cup s^d}(C) + 4K \cdot W_N(\hat{\nu}_{s \cup s^d}, \hat{\nu}_{t \cup t^d}) + \eta^* + 4K \cdot \sqrt{\frac{2}{\eta'} \log\left(\frac{1}{\delta}\right)} \left(\sqrt{\frac{1}{N_s}} + \sqrt{\frac{1}{N_t}}\right)$$

## C  CONVERGENCE ANALYSIS OF D-NWD

In Figure 3, we plot a training curve showing the accuracy and D-NWD loss across epochs to monitor the model's performance during training. The left plot represents accuracy over epochs, where the model's accuracy improves and fluctuates around $52\text{-}55\%$, considered a strong performance for this specific dataset. The early improvement followed by consistent performance indicates that the model is learning to differentiate between classes or features as training progresses and later gradually converges in terms of accuracy. On the right, the D-NWD loss plot shows a continuous decrease as training progresses. This indicates that the model is consistently optimizing its objective. As training continues, the accuracy improvements slow down, and the loss converges toward a stable value.

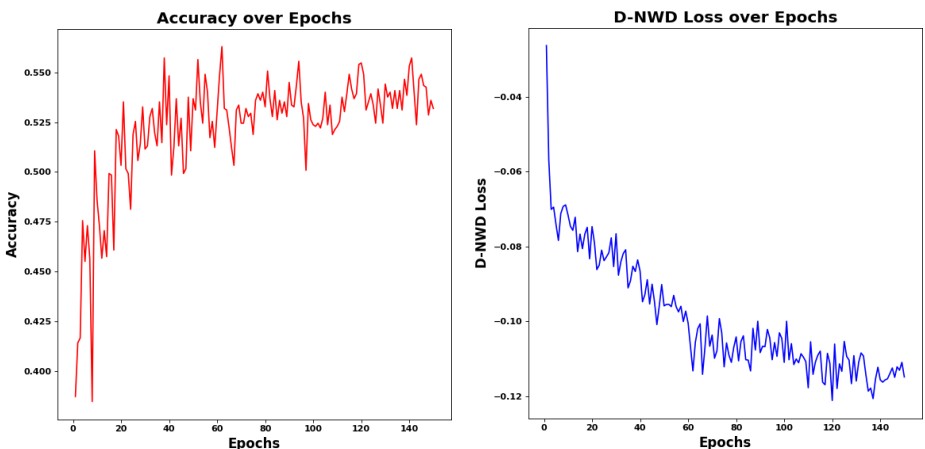

Figure 3: Trianing curve for ScanNet to ShapeNet task from PointDA.

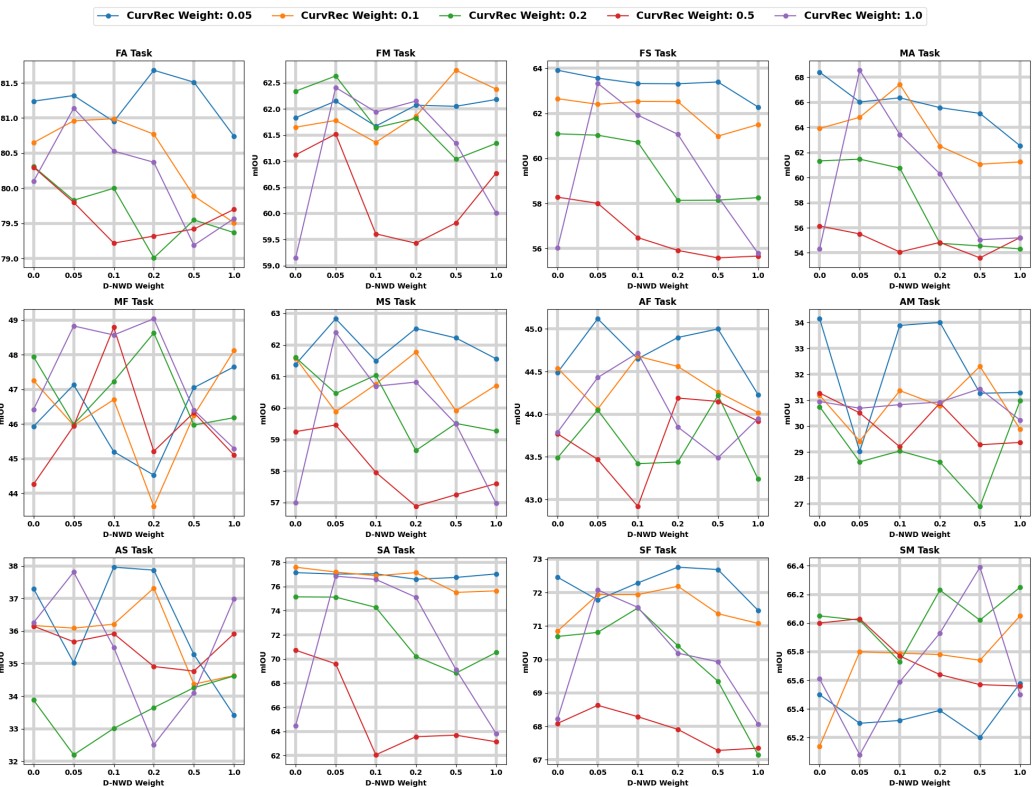

Figure 4: The performance of CDND with different hyperparameter settings on PointSegDA dataset.

# D EXPERIMENT DETAILS

**Implementaion Details.** Our code is based on the open-source implementation of the De-fRec+PCM. We trained our three CDND models with seeds $\{1, 2, 3\}$ on A100 GPUs. For the PointSegDA dataset, we fixed the learning rate to be 0.001 and conducted a grid search to opti-mize the hyperparameters $\alpha$, $\gamma$, $\beta_1$, and $\beta_2$ for each task. The specific hyperparameter values can be

found in Table 4. Similarly, for the PointDA dataset, the hyperparameters are listed in Table 5. The training time for tasks in the PointDA dataset is approximately 10 hours, resulting in a high computational cost for hyperparameter tuning. Therefore, we do not tune the hyperparameters extensively. Similarly, for GAST and ImplicitPCDA, we use the hyperparameters provided in their open-source code (GAST, ImplicitPCDA) for the PointDA dataset.

However, GAST and ImplicitPCDA have not been tested on the PointSegDA dataset before. When implementing GAST, we conduct a grid search on the PointSegDA dataset, exploring values of 0.1, 0.2, 0.5, and 1.0 for both $\mathcal{L}_{rot}$ and $\mathcal{L}_{loc}$. For ImplicitPCDA, we perform a grid search on the PointSegDA dataset, considering values of 0.1, 0.2, 0.5, and 1.0 for $\mathcal{L}_M$. Please refer to the original papers Zou et al. (2021); Shen et al. (2022) for the definitions of $\mathcal{L}_{rot}$, $\mathcal{L}_{loc}$, and $\mathcal{L}_M$.

Table 4: Hyperparameters for PointSegDA.

| Hyperparameter | Values |
|---|---|
| Learning Rate | 0.001 |
| $\alpha$ | 1.0 |
| $\gamma$ | [0.05, 0.1, 0.2, 0.5, 1.0] |
| $\beta_1$ | [0.0, 0.05, 0.1, 0.2, 0.5, 1.0] |
| $\beta_2$ | [0.0, 0.2] |

Table 5: Hyperparameters for PointDA.

| Hyperparameter | Values |
|---|---|
| Learning Rate | 0.001, 0.0001 (S$^+$M, MS) |
| $\alpha$ | 0.5 |
| $\gamma$ | 0.5 |
| $\beta_1$ | [0.0, 1.0] |
| $\beta_2$ | 0.2 |

**Challenges of Applying SPST with mIoU.** The mIOU metric is defined as:

$$\text{mIoU} = \frac{1}{M} \sum_{m=1}^{M} \frac{TP_m}{TP_m + FP_m + FN_m} \quad \text{where:} \quad \begin{aligned} M &= \text{number of classes} \\ TP_m &= \text{true positive for class } m \\ FP_m &= \text{false positive for class } m \\ FN_m &= \text{false negative for class } m \end{aligned}$$

SPST typically relies on ranking training samples by difficulty and gradually incorporating harder examples into training. The training samples for point cloud segmentation tasks are points in point clouds. However, mIoU is a global metric that evaluates performance across an entire point cloud, making it challenging to assign difficulty scores to individual points in a point cloud. The mechanism of SPST mismatches the per-point cloud, rather than per-point, evaluation criterion of mIoU.

# E HYPERPARAMETER SENSITIVITY ANALYSIS

We perform a sensitivity analysis on the PointSegDA dataset. The results are shown in Figure 4. We select our hyperparameters based on the model's performance on the validation set of the target domain dataset. We focus on the hyperparameters with the most values for selection: Curvature diversity-based deformation reconstruction (CurvRec(En)-Low) and D-NWD. For the CurvRec weight ($\gamma$), we search across 0.05, 0.1, 0.2, 0.5, and 1.0. For the D-NWD weight ($\beta_1$), we explore 0.0, 0.05, 0.1, 0.2, 0.5, and 1.0.

The tasks FA, FM, MF, AF, and SM exhibit relative insensitivity to changes in the CurvRec and D-NWD weights, as their mIOU values tend to fluctuate within a narrow range, typically around 5%. This suggests that these tasks are more robust to variations in these hyperparameters compared to others. In general, higher CurvRec weights tend to result in worse performance, particularly for tasks like FS, MA, and SA, where we observe a drop in mIOU as CurvRec weight increases. Higher D-NWD weights (*e.g.*, 1.0) generally have a negative impact, leading to a decline in mIOU for many tasks, including FS, MA, and SF. Lower D-NWD weights (*e.g.*, 0.0 or 0.05) are also generally associated with worse mIOU performance across tasks like MF, MS, and AS, suggesting that a lighter emphasis on D-NWD is disadvantageous in these cases. However, moderate D-NWD weights (*e.g.*, 0.1, 0.2, 0.5) lead to better results, as shown in the MS, AF, AM, SF, SA, and AS tasks. In summary, a higher CurvRec weight combined with a moderate D-NWD weight enhances model performance.

