# OpenReview forum: "Curvature Diversity-Driven Deformation and Domain Alignment for Point Cloud"
_ICLR.cc/2025/Conference — ICLR 2025 Conference Withdrawn Submission_

### Official Review · Reviewer_wK5w · 2024-10-31

**Soundness:** 2
**Presentation:** 2
**Contribution:** 1
**Rating:** 5
**Confidence:** 4

**Summary:**

This paper presents the Curvature Diversity-driven Deformation Reconstruction (CurvRec) task and the Deformation-based Nuclear-norm Wasserstein Discrepancy (D-NWD) for aligning source and target domains. Theoretical analysis confirms the generality of D-NWD, while experiments on two public domain adaptation datasets demonstrate the effectiveness of the proposed approach.

**Strengths:**

1.	CDND guides the feature extractor to focus on semantically rich regions.
2.	D-NWD effectively integrates features from both original and deformed samples to align the source and target domains.
3.	This approach achieves state-of-the-art performance.

**Weaknesses:**

1.	Lack of novelty. Several key components in the proposed method are direct extensions of prior work. For example, GRL was introduced in the 2014 paper "Unsupervised Domain Adaptation by Backpropagation" and has been extensively studied in subsequent domain adaptation research. Additionally, using Chamfer Distance for reconstruction as a self-supervised loss is a standard practice in the point cloud domain.
2.	Outdated baselines. The method is primarily compared with the 2021 ImplicitPCDA, which is insufficient to demonstrate the superiority of CDND. Please include comparisons with more recent approaches.
3.	Missing citations. Add citations following the method names in the main experimental table, and consider comparing the proposed method with more cutting-edge work.

**Questions:**

Please refer to Weakness

---

### Official Review · Reviewer_AAAk · 2024-11-03

**Soundness:** 3
**Presentation:** 4
**Contribution:** 2
**Rating:** 5
**Confidence:** 4

**Summary:**

This paper proposes a method for domain alignment of point clouds called Curvature Diversity-Driven Nuclear-Norm Wasserstein Domain Alignment (CDND). It proposes a deformation reconstruction method that leverages curvature diversity in different regions of a point cloud for domain alignment. Regions are selected for deformation based on entropy, avoiding semantically rich regions. It then proposes a Deformation-based Nuclear-norm Wasserstein Discrepancy (D-NWD) method that incorporates features from the original and deformed samples for improved domain alignment, theoretically analysing D-NWD to show that it reduces domain gap and generalizes to any deformation type.  Results are provided on classification and segmentation benchmarks.

**Strengths:**

The paper tackles an important problem in point cloud classification and segmentation. The paper seems technically correct and is well written and easy to follow.

State of the art results are reported for domain adaptation for point cloud classification and segmentation tasks.

**Weaknesses:**

The novelty of the proposed method is limited because  :

(1) Deformation reconstruction for domain adaptation of point clouds was already proposed by Achituve et al. (CVPR 2021). The main difference of this paper is that it proposes a different method for selecting the regions for deformation i.e. “deforming regions that are less semantically rich”.

(2) Wasserstein distance is commonly used for domain alignment and the authors explicitly mention that their work is inspired by the Nuclear-norm Wasserstein discrepancy (NWD) by Chen et al. 2022. The main difference in this paper is that it additionally considers features from the deformed samples as well. Including the deformed samples for learning the domain shift by D-NWD is simply a method of data augmentation.

Hence, the novelty is incremental in my opinion. Moreover, I don’t think replacing the selected regions (for deformation) with Gaussian noise (lines 194-195) represents variations in real world data.


Typos:

Sometimes references are repeated e.g. “Qin et al. Qin et al. (2019)” and “Achituve et al. Achituve et al. (2021)” and sometimes they are part of the sentence when they shouldn’t be e.g. “use Farthest Point Sampling (FPS) Moenning & Dodgson (2003)”. Authors should check and correct all.

“three three domains” -> three domains

Too many bold letters in the text are somewhat distracting.

**Questions:**

See my comments above.

---

### Official Review · Reviewer_PLpe · 2024-11-08

**Soundness:** 3
**Presentation:** 1
**Contribution:** 2
**Rating:** 3
**Confidence:** 3

**Summary:**

This paper proposes a solution for unsupervised domain adaptation for 3D point clouds. It uses a curvature diversity-driven deformation reconstruction task  to reduce the domain gap between the source and target domain，which enables the model to extract salient features from semantically rich regions of a point cloud. And it utilizes a Deformation-based Nuclear-norm Wasserstein Discrepancy to align the source and target domains. Experiments on point cloud classification and segmentation tasks  are conducted to validate the proposed method.

**Strengths:**

1. The paper provides theoretical justification for the proposed method.
2. The proposed method shows clear gains on both PointDA-10 dataset and PointSegDA dataset.

**Weaknesses:**

The overall writing quality is quite low.
1. The motivation of this work is unclear. As the authors state in the abstract, a significant challenge of UDA lies in effectively bridging the domain gap, but this applies for any data representation. What is the specialty of unsupervised domain adaptation for 3D point cloud learning?
2. The definition of entropy of curvature is heuristic and needs more solid justification. What is the definition of the j-th region of a point cloud? Why $10^{-10}$, not another constant?  It is not clear how the entropy of curvature is applied in the process of curvature diversity-driven deformation ( Lines 189-196).
3. The D-NWD objective is not well-defined. Many notations are not explained. What are $v_s$ and $v_t$? The only difference between NWD  and D-NWD is the appearance of $v_t$? This seems quite trivial.
4. In the overall loss (10), why still need $L_{NWD}$, given $L_{D-NWD}$?
5. The theoretical results are not well explained and very hard to follow . There is no explanation about Theorem 1 at all. It is impossible to recognize that why D-NWD can outperform NWD from the proposed theoretical results.
6. The baseline methods in the comparison are relatively dated. The authors should compare with more recent methods (works published in 2023, 2024).
7. There is no ablation study about the hyper-parameters, like $\alpha, \gamma, \beta_1, \beta_2$.
8. Table 2 is hard to understand. What does each model mean? What does Low in CurvRec(S)-Low mean?
9. The description about the future work has nothing to do with 3D point clouds, which seems not quite sloppy.

**Questions:**

The questions are included in the weakness section.

---

### Note · Authors · 2024-11-12

I have read and agree with the venue's withdrawal policy on behalf of myself and my co-authors.